# On the Power and Limitations of Random Features for Understanding Neural Networks

**Gilad Yehudai** **Ohad Shamir**
Weizmann Institute of Science
{gilad.yehudai,ohad.shamir}@weizmann.ac.il

## Abstract

Recently, a spate of papers have provided positive theoretical results for training over-parameterized neural networks (where the network size is larger than what is needed to achieve low error). The key insight is that with sufficient over-parameterization, gradient-based methods will implicitly leave some components of the network relatively unchanged, so the optimization dynamics will behave as if those components are essentially fixed at their initial random values. In fact, fixing these *explicitly* leads to the well-known approach of learning with random features (e.g. [27, 29]). In other words, these techniques imply that we can successfully learn with neural networks, whenever we can successfully learn with random features. In this paper, we formalize the link between existing results and random features, and argue that despite the impressive positive results, random feature approaches are also inherently limited in what they can explain. In particular, we prove that random features cannot be used to learn *even a single ReLU neuron* (over standard Gaussian inputs in $\mathbb{R}^d$ and poly($d$) weights), unless the network size (or magnitude of its weights) is exponentially large in $d$. Since a single neuron *is* known to be learnable with gradient-based methods, we conclude that we are still far from a satisfying general explanation for the empirical success of neural networks. For completeness we also provide a simple self-contained proof, using a random features technique, that one-hidden-layer neural networks can learn low-degree polynomials.

## 1 Introduction

Deep learning, in the form of artificial neural networks, has seen a dramatic resurgence in popularity in recent years. This is mainly due to impressive performance gains on various difficult learning problems, in fields such as computer vision, natural language processing and many others. Despite the practical success of neural networks, our theoretical understanding of them is still very incomplete.

A key aspect of modern networks is that they tend to be very large, usually with many more parameters than the size of the training data: In fact, so many that in principle, they can simply memorize all the training examples (as shown in the influential work of Zhang et al. [40]). The fact that such huge, over-parameterized networks are still able to learn and generalize is one of the big mysteries concerning deep learning. A current leading hypothesis is that over-parameterization makes the optimization landscape more benign, and encourages standard gradient-based training methods to find weight configurations that fit the training data as well as generalize (even though there might be many other configurations which fit the training data without any generalization). However, pinpointing the exact mechanism by which over-parameterization helps is still an open problem.

Recently, a spate of papers (such as [4, 11, 14, 2, 23, 15, 9, 3, 1]) provided positive results for training and learning with over-parameterized neural networks. Although they differ in details, they are all based on the following striking observation: When the networks are sufficiently large, standard

gradient-based methods change certain components of the network (such as the weights of a certain layer) very slowly, so that if we run these methods for a bounded number of iterations, they might as well be fixed. To give a concrete example, consider one-hidden-layer neural networks, which can be written as a linear combination of $r$ neurons

$$N(x) = \sum_{i=1}^{r} u_i \sigma(\langle w_i, x \rangle + b_i) , \tag{1}$$

using weights $\{u_i, w_i, b_i\}_{i=1}^{r}$ and an activation function $\sigma$. When $r$ is sufficiently large, and with standard random initializations, it can be shown that gradient descent will leave the weights $w_i, b_i$ in the first layer nearly unchanged (at least initially). As a result, the dynamics of gradient descent will resemble those where $\{w_i, b_i\}$ are fixed at random initial values – namely, where we learn a *linear* predictor (parameterized by $u_1, \ldots, u_r$) over a set of $r$ random features of the form $x \mapsto \sigma(\langle w_i, x \rangle + b_i)$ (for some random choice of $w_i, b_i$). For such linear predictors, it is not difficult to show that they will converge quickly to an optimal predictor (over the span of the random features). This leads to learning guarantees with respect to hypothesis classes which can be captured well by such random features: For example, most papers focus (explicitly or implicitly) on multivariate polynomials with certain constraints on their degree or the magnitude of their coefficients. We discuss these results in more detail (and demonstrate their close connection to random features) in Section 2.

Taken together, these results are a significant and insightful advance in our understanding of neural networks: They rigorously establish that sufficient over-parameterization allows us to learn complicated functions, while solving a non-convex optimization problem. However, it is important to realize that this approach can only explain learnability of hypothesis classes which can already be learned using random features. Considering the one-hidden-layer example above, this corresponds to learning linear predictors over a fixed representation (chosen obliviously and randomly at initialization). Thus, it does not capture any element of *representation learning*, which appears to lend much of the power of modern neural networks.

In this paper we show that there are inherent limitations on what predictors can be captured with random features, and as a result, on what can be provably learned with neural networks using the techniques described earlier. We consider features of the form $f_i : \mathbb{R}^d \to \mathbb{R}$ which are chosen (randomly or deterministically) and then fixed. The $f_i$'s can be arbitrary functions, including multilayered neural networks, as long as their norm (suitably defined) is not exponential in the input dimension. We show that using $N(x) = \sum_{i=1}^{r} u_i f_i(x)$ we cannot efficiently approximate *even a single ReLU neuron*:

**Theorem 1.1** (Informal version of Theorem 4.8). *Let $\mathcal{F}$ be a family of function on $\mathbb{R}^d$, where for every $f \in \mathcal{F}$, $\mathbb{E}_{x \sim \mathcal{N}(0,I)}[f(x)^2]$ is less than exponential in the dimension $d$, and let $D$ be a distribution over tuples $(f_1, \ldots, f_r)$ of functions from $\mathcal{F}$. Then there exist a weight vector $w^* \in \mathbb{R}^d$, $\|w^*\| = d^2$ and bias term $b^* \in \mathbb{R}$ such that w.h.p over the choice of functions from $\mathcal{F}$, if:*

$$\mathbb{E}_{x \sim \mathcal{N}(0,I)} \left[ \left( N(x) - [\langle w^*, x \rangle + b^*]_+ \right)^2 \right] \leq \frac{1}{50}$$

*(where $[z]_+ = \max\{0, z\}$ is the ReLU function and $x$ has a standard Gaussian distribution) then*

$$r \cdot \max_i |u_i| \geq \exp(\Omega(d)).$$

In other words, either the number of neurons $r$ or the magnitude of the weights (or both) must be exponential in the dimension $d$, which generally implies exponential time to learn the target function (see Remark 4.5). Moreover, if the features can be written as functions operating on a random linear transformation over the data, then the same result holds for any choice of $w^*$. In more details:

**Theorem 1.2** (Informal version of Theorem 4.2). *Let $\mathcal{F}$ be a family of functions where each $f \in \mathcal{F}$ can be written as $x \mapsto f(Wx)$ where $W$ is a random matrix whose rows are sampled uniformly at random from the unit sphere. Then the result of theorem 3.1 holds for any $w^* \in \mathbb{R}^d$, $\|w^*\| = d^2$.*

Both theorems apply for a large family of functions, particular examples include two layered neural networks (Theorm 1.2) and "linearized" neural tangent kernel (Theorem 1.1, also see Jacot et al. [20]). These results imply that the random features approach cannot fully explain polynomial-time learnability of neural networks, even with respect to data generated by an extremely simple neural network, composed of a single neuron. This is despite the fact that single ReLU neurons *are* easily

learnable with gradient-based methods (e.g., [26], [34], also see Section 4 for further details). The point we want to make here is that the random feature approach, as a theory for explaining the success of neural networks, cannot explain even the fact that single neurons are learnable.

For completeness we also provide a simple, self-contained analysis, showing how over-parameterized, one-hidden-layer networks can provably learn polynomials with bounded degrees and coefficients, using standard stochastic gradient descent with standard initialization.

We emphasize that there is no contradiction between our positive and negative results: In the positive result on learning polynomials, the required size of the network is exponential in the *degree* of the polynomial, and low-degree polynomials cannot express even a single ReLU neuron if its weights are large enough.

Overall, we argue that although the random feature approach captures important aspects of training neural networks, it is by no means the whole story, and we are still quite far from a satisfying general explanation for the empirical success of neural networks.

### Related Work

The recent literature on the theory of deep learning is too large to be thoroughly described here. Instead, we survey here some of the works most directly relevant to the themes of our paper. In Section 2, we provide a more technical explanation on the connection of recent results to random features.

**The Power of Over-Parameterization.** The fact that over-parameterized networks are easier to train was empirically observed, for example, in [25], and was used in several contexts to show positive results for learning and training neural networks. For example, it is known that adding more neurons makes the optimization landscape more benign (e.g., [30, 36, 37, 31, 10, 35]), or allows them to learn in various settings (e.g., besides the papers mentioned in the introduction, [8, 24, 7, 39, 13]).

**Random Features.** The technique of random features was proposed and formally analyzed in [27, 28, 29], originally as a computationally-efficient alternative to kernel methods (although as a heuristic, it can be traced back to the "random connections" feature of Rosenblatt's Perceptron machine in the 1950's). These involve learning predictors of the form $x \mapsto \sum_{i=1}^{r} u_i \psi_i(x)$, where $\psi_i$ are random non-linear functions. The training involves only tuning of the $u_i$ weights. Thus, the learning problem is as computationally easy as training linear predictors, but with the advantage that the resulting predictor is non-linear, and in fact, if $r$ is large enough, can capture arbitrarily complex functions. The power of random features to express certain classes of functions has been studied in past years (for example [5, 28, 21, 38]). However, in our paper we also consider negative rather than positive results for such features. [5] also discusses the limitation of approximating functions with a bounded number of such features, but in a different setting than ours (worst-case approximation of a large function class using a fixed set of features, rather than inapproximability of a fixed target function, and not in the context of single neurons). Less directly related, [41] studied learning neural networks using kernel methods, which can be seen as learning a linear predictor over a fixed non-linear mapping. However, the algorithm is not based on training neural networks with standard gradient-based methods. In a very recent work (and following the initial dissemination of our paper), Ghorbani et al. [17] studied the representation capabilities of random features, and showed that in high dimensions random features are not good at fitting high degree polynomials.

### Notation

Denote by $U\left([a, b]^d\right)$ the $d$-dimensional uniform distribution over the rectangle $[a, b]^d$, and by $\mathcal{N}(0, \Sigma)$ the multivariate Gaussian distribution with covariance matrix $\Sigma$. For $T \in \mathbb{N}$ let $[T] = \{1, 2, \ldots, T\}$, and for a vector $w \in \mathbb{R}^d$ we denote by $\|w\|$ the $L_2$ norm. We denote the ReLU function by $[x]_+ = \max\{0, x\}$.

## 2 Analysis of Neural Networks as Random Features

In many previous works, a key element in the analysis of neural networks is to build a reduction from neural networks to random features. In this reduction, usually by choosing appropriate learning rate and initialization, close to the initialization gradient descent is optimizing neural network similarly to

the way it would optimize random features. Thus, it is enough to analyze the optimization process of random features, and deduce that if random features can achieve good performance, the same holds for neural networks. Here we survey some of these works and how they can actually be viewed as random features.

## 2.1 Optimization with Coupling, Fixing the Output Layer

One approach is to fix the output layer and do optimization only on the inner layers. Most works that use this method (e.g. [23], [15], [3], [9], [2], [1]) also use the method of "coupling" and the popular ReLU activation. This method uses the following observation: a ReLU neuron can be viewed as a linear predictor multiplied by a threshold function, that is: $[\langle w, x \rangle]_+ = \langle w, x \rangle \mathbb{1}_{\langle w, x \rangle \geq 0}$. The coupling method informally states that after doing gradient descent with appropriate learning rate and a limited number of iterations, the amount of neurons that change the sign of $\langle w, x \rangle$ (for $x$ in the data) is small. Thus, it is enough to analyze a linear network over random features of the form: $x \mapsto \langle w, x \rangle \mathbb{1}_{\langle w^{(0)}, x \rangle \geq 0}$ where $w^{(0)}$ are randomly chosen.

For example, a one-hidden-layer neural network where the activation $\sigma$ is the ReLU function can be written as

$$\sum_{i=1}^{r} u_i^{(t)} \sigma(\langle w_i^{(t)}, x \rangle) = \sum_{i=1}^{r} u_i^{(t)} \langle w_i^{(t)}, x \rangle \mathbb{1}_{\langle w_i^{(t)}, x \rangle \geq 0}.$$

Using the coupling method, after doing gradient descent, the amount of neurons that change sign, i.e. the sign of $\langle w_i^{(t)}, x \rangle$ changes, is small. As a result, using the homogeneity of the ReLU function, the following network can actually be analyzed:

$$\sum_{i=1}^{r} u_i^{(t)} \langle w_i^{(t)}, x \rangle \mathbb{1}_{\langle w_i^{(0)}, x \rangle \geq 0} = \sum_{i=1}^{r} \langle u_i^{(t)} \cdot w_i^{(t)}, x \rangle \mathbb{1}_{\langle w_i^{(0)}, x \rangle \geq 0},$$

where $w_i^{(0)}$ are randomly chosen. This is just analyzing a linear predictor with random features of the form $x \mapsto x_j \mathbb{1}_{\langle w_i^{(0)}, x \rangle \geq 0}$. Note that the homogeneity of the ReLU function is used in order to show that fixing the output layer does not change the network's expressiveness. This is not true in terms of optimization, as optimizing both the inner layers and the output layers may help the network converge faster, and to find a predictor which has better generalization properties. Thus, the challenge in this approach is to find functions or distributions that can be approximated with this kind of random features network, using a polynomial number of features.

## 2.2 Optimization on all the Layers

A second approach in the literature (e.g. Andoni et al. [4], Daniely et al. [12], Du et al. [14]) is to perform optimization on all the layers of the network, choose a "good" learning rate and bound the number of iterations such that the inner layers stay close to their initialization. For example, in the setting of a one-hidden-layer network, for every $\epsilon > 0$, a learning rate $\eta$ and number of iterations $T$ are chosen, such that after running gradient descent with these parameters, there is an iteration $1 \leq t \leq T$ such that:

$$\left\| U^{(t)} \sigma(W^{(t)} x) - U^{(t)} \sigma(W^{(0)} x) \right\| \leq \epsilon.$$

Hence, it is enough to analyze a linear predictor over a set of random features:

$$U^{(t)} \sigma\left(W^{(0)} x\right) = \sum_{i=1}^{r} u_i^{(t)} \sigma\left(\langle w_i^{(0)}, x \rangle\right),$$

where $\sigma$ is not necessarily the ReLU function. Again, the difficulty here is finding the functions that can be approximated in this form, where $r$ (the amount of neurons) is only polynomial in the relevant parameters.

## 3 Over-Parameterized Neural Networks Learn Polynomials

For completeness we provide a simple, self-contained analysis, showing how over-parameterized, one-hidden-layer networks can provably learn polynomials with bounded degrees and coefficients, using standard stochastic gradient descent with standard initialization. In more details:

**Theorem 3.1** (Informal). *Given any distribution $D$ over labeled data $(x, y)$, where $x \in \mathbb{R}^d$, $y \in \{-1, +1\}$, any $\epsilon > 0$, and almost any multivariate polynomial $P(x)$ with degree at most $k$ and coefficients of magnitude at most $\alpha$, if we take a one-hidden-layer neural network $N(x)$ with analytic activation functions which are $L - Lipschitz$, and with at least*

$$r > poly\left(\frac{1}{\epsilon}, \log\left(\frac{1}{\delta}\right), d^{k^2}, \alpha^k, L\right)$$

*neurons, and run $poly(r)$ many iterations of stochastic gradient descent on i.i.d. examples, then with probability at least $1 - \delta$ over the random initialization, it holds that*

$$\mathbb{E}[L_D(N(x))] \leq L_D(P(x)) + \epsilon,$$

*where $L_D$ is the expected hinge loss and the expectation is over the random choice of examples.*

For a formal statement and full proof see Appendix C. We emphasize that although our analysis improves on previous ones in certain aspects (discussed in more detail in Appendix A), it is not fundamentally novel: Our goal is mostly to present a transparent and self-contained result using the approach developed in previous papers, focusing on clarity rather than generality. In comparison, some of the related results assume that the output layer is fixed (although in practice all layers are optimized); some focus on training error rather than population risk; some do not quantitatively characterize the class of polynomials learned by the network; and some consider different networks architectures or optimization methods. For an in-depth review of previous works and how they differ from the above result see Appendix A.

## 4 Limitations of Random Features

Having discussed and shown positive results for learning using (essentially) random features, we turn to discuss the limitations of this approach.

Concretely, we will consider in this section data $(x, y)$, where $x \in \mathbb{R}^d$ is drawn from a standard Gaussian on $\mathbb{R}^d$, and there exists some single ground-truth neuron which generates the target values $y$: Namely, $y = \sigma(\langle w^*, x \rangle + b^*)$ for some fixed $w^*, b^*$. We also consider the squared loss $l(\hat{y}, y) = (\hat{y} - y)^2$, so the expected loss we wish to minimize takes the form

$$\mathbb{E}_x\left[\left(\sum_{i=1}^{r} u_i f_i(x) - \sigma(\langle w^*, x \rangle + b^*)\right)^2\right] \tag{2}$$

where $f_i$ are the random features. Importantly, when $r = 1$, $\sigma$ is the ReLU function, and $f_1(x) = \sigma(\langle w, x \rangle + b)$ (that is, we train a single neuron to learn a single target neuron), this problem *is* quite tractable with standard gradient-based methods (see, e.g., [26], [34]). In this section, we ask whether this positive result – that single target neurons can be learned – can be explained by the random features approach. Specifically, we consider the case where the function $f_i$ are arbitrary functions chosen obliviously of the target neuron (e.g. multilayered neural networks at a standard random initialization), and ask what conditions on $r$ and $u_i$ are required to minimize Eq. (2). Our main results (Theorem 4.2 and Theorem 4.8) show that either one of them has to be exponential in the dimension $d$, as long as the sizes of $w^*, b^*$ are allowed to be polynomial in $d$. Since networks with exponentially-many neurons or exponentially-sized weights are generally not efficiently trainable, we conclude that an approach based on random-features cannot explain why learning single neurons is tractable in practice. In Theorem 4.2, we show the result for *any* choice of $w^*$ with some fixed norm, but require the feature functions to have a certain structure (which is satisfied by neural networks). In Theorem 4.8, we drop this requirement, but then the result only holds for a particular $w^*$.

To simplify the notation in this section, we consider functions on $x$ as elements of the $L^2(\mathbb{R}^d)$ space weighted by a standard Gaussian measureSpecifically, for a function $f : \mathbb{R}^d \to \mathbb{R}$ we denote

$$\|f(\cdot)\|^2 := \mathbb{E}_{x \sim \mathcal{N}(0, I)}\left[f^2(x)\right] = c_d \int_{\mathbb{R}^d} f^2(x) e^{\frac{-\|x\|^2}{2}} dx,$$

$$\langle f(\cdot), g(\cdot) \rangle := \mathbb{E}_{x \sim \mathcal{N}(0, I)}[f(x)g(x)] = c_d \int_{\mathbb{R}^d} f(x)g(x) e^{\frac{-\|x\|^2}{2}} dx,$$

where $c_d = \left(\frac{1}{\sqrt{2\pi}}\right)^d$ is a normalization term. For example, Eq. (2) can also be written as $\|\sum_{i=1}^{r} u_i f_i(\cdot) - \sigma(\langle w^*, \cdot \rangle + b^*)\|^2$.

## 4.1 Warm up: Linear predictors

Before stating our main results, let us consider a particularly simple case, where $\sigma$ is the identity, and our goal is to learn a *linear predictor* $x \mapsto \langle w^*, x \rangle$ with $\|w^*\| = 1$. We will show that already in this case, there is a significant cost to pay for using random features. The main result in the next subsection can be seen as an elaboration of this idea.

In this setting, finding a good linear predictor, namely minimizing $\|\langle w, \cdot \rangle - \langle w^*, \cdot \rangle\|$ is easy: It is a convex optimization problem, and is easily solved using standard gradient-based methods. Suppose now that we are given random features $w_1, \ldots, w_r \sim N\left(0, \frac{1}{d}I_d\right)$ and want to find $u_1, \ldots, u_r \in \mathbb{R}$ such that:

$$\left\| \sum_{i=1}^{r} u_i \langle w_i, \cdot \rangle - \langle w^*, \cdot \rangle \right\| \leq \epsilon. \tag{3}$$

The following proposition shows that with high probability, Eq. (3) cannot hold unless $r = \Omega(d)$. This shows that even for linear predictors, there is a price to pay for using a combination of random features, instead of learning the linear predictor directly.

**Proposition 4.1.** *Let $w^*$ be some unit vector in $\mathbb{R}^d$, and suppose that we pick random features $w_1, \ldots, w_r$ i.i.d. from any spherically symmetric distribution in $\mathbb{R}^d$. If $r \leq \frac{d}{2}$, then with probability at least $1 - \exp(-cd)$ (for some universal constant $c > 0$), for any choice of weights $u_1, \ldots, u_r$, it holds that $\left\| \sum_{i=1}^{r} u_i \langle w_i, \cdot \rangle - \langle w^*, \cdot \rangle \right\|^2 \geq \frac{1}{4}$.*

The full proof appears in Appendix B, but the intuition is quite simple: With random features, we are forced to learn a linear predictor in the span of $w_1, \ldots, w_r$, which is a random $r$-dimensional subspace of $\mathbb{R}^d$. Since this subspace is chosen obliviously of $w^*$, and $r \leq \frac{d}{2}$, it is very likely that $w^*$ is not close to this subspace (namely, the component of $w^*$ orthogonal to this subspace is large), and therefore we cannot approximate this target linear predictor very well.

## 4.2 Features Based on Random Linear Transformations

Having discussed the linear case, let us return to the case of a non-linear neuron. Specifically, we will show that even a single ReLU neuron cannot be approximated by a very large class of random feature predictors, unless the amount of neurons in the network is exponential in the dimension, or the coefficients of the linear combination are exponential in the dimension. In more details:

**Theorem 4.2.** *There exists a universal constant $c > 0$ such that the following holds. Let $d > 40$, $k \in \mathbb{N}$, and let $\mathcal{F}$ be a family of functions from $\mathbb{R}^k$ to $\mathbb{R}$. Also, let $W \in \mathbb{R}^{k \times d}$ be a random matrix whose rows are sampled uniformly at random from the unit sphere. Suppose that $f_W(x) := f(Wx)$ satisfies $\|f_W(\cdot)\| \leq \exp(d/40)$ for any realization of $W$ and for all $f \in \mathcal{F}$. Then for every $w^* \in \mathbb{R}^d$ with $\|w^*\| = d^2$, there exists $b^* \in \mathbb{R}$ with $|b^*| \leq 6d^3 + 1$, such that for any $f_1, \ldots f_r \in \mathcal{F}$, w.p $> 1 - \exp(-cd)$ over sampling $W$, if*

$$\mathbb{E}_{x \sim \mathcal{N}(0,I)} \left[ \left( \sum_{i=1}^{r} u_i f_i(Wx) - [\langle w^*, x \rangle + b^*]_+ \right)^2 \right] \leq \frac{1}{50},$$

*then*

$$r \cdot \max_i |u_i| \geq \frac{1}{48d^2} \exp(cd).$$

Note that the theorem allows any "random" feature which can be written as a composition of some function $f$ (chosen randomly or not), and a random linear transformation.

**Example 4.3.** *As a special case of Theorem 4.2, consider two layer neural networks with activation $\sigma : \mathbb{R} \to \mathbb{R}$ and without a bias term. Denote by $W_i$ the $i$-th row of $W$, then two layer neural networks can be viewed as a linear combination of functions of the form:*

$$f_i(Wx) = \sigma(\langle W_i, x \rangle),$$

*where if each $W_i$ is bounded and the norm of the activation $\sigma$ is bounded, then the norm of $f_i(Wx)$ is also bounded. This example can be extended to multi-layer neurons of any depth as long as the first layer performs a random linear transformation.*

**Remark 4.4.** *The requirement that the rows of $W$ are sampled uniformly from the unit sphere can be easily relaxed so that the rows of $W$ have a spherically symmetrical distribution and are bounded w.h.p. These kind of distributions includes, among others, bounded distributions and standard Gaussian distributions. However it would require a more careful analysis on the bound of the functions $f_W$, as they might be only bounded w.h.p.*

**Remark 4.5.** *As opposed to the linear case, here we also have a restriction on $\max_i |u_i|$. We conjecture that it is possible to remove this dependence and leave it to future work. With that said, existing analyses of stochastic gradient descent, even for convex functions, imply that the required number of iterations scales polynomially with the norm of the target solution (e.g., Hazan et al. [19]), which would mean exponentially many iterations in our case. Moreover, practically speaking, such huge coefficients can cause overflow when running SGD on a computer with standard floating point formats.*

To prove the theorem, we will use the following proposition, which implies that functions of the form $x \mapsto \psi(\langle w, x \rangle)$ for a certain sine-like $\psi$ and "random" $w$ are nearly uncorrelated with any fixed function.

**Proposition 4.6.** *Let $d \in \mathbb{N}$, where $d > 40$, and let $a = 6d^2 + 1$. Define the following function:*

$$\psi(x) = [x + a]_+ + \sum_{n=1}^{a} 2[x + a - 2n]_+ (-1)^n - 1,$$

*Then $\psi : \mathbb{R} \to \mathbb{R}$ satisfies the following:*

1. *It is a periodic odd function on the interval $[-a, a]$*

2. *For every $w^* \in \mathbb{R}^d$ with $\|w^*\| = d$, $\|\psi(\langle w^*, \cdot \rangle)\|^2 \geq d$.*

3. *For every $f \in L^2(\mathbb{R}^d)$, we have $\mathbb{E}_w \left( \langle f(\cdot), \psi(\langle w, \cdot \rangle) \rangle^2 \right) \leq 20\|f\|^2 \cdot \exp(-cd)$, where $w$ is sampled uniformly from $\{w : \|w\| = d\}$, and $c > 0$ is a universal constant.*

Items 1 and 2 follow by a straightforward calculation, where in item 2 we also used the fact that $x$ has a symmetric distribution. Item 3 relies on a claim from [33], which shows that periodic functions of the form $x \mapsto \psi(\langle w, x \rangle)$ for a random $w$ with sufficiently large norm have low correlation with any fixed function. The full proof can be found in Appendix B.

At a high level, the proof of Theorem 4.2 proceeds as follows: If we choose and fix $\{f_i\}_{i=1}^r$ and $W$, then any linear combination of random features $f_i(Wx)$ with small weights will be nearly uncorrelated with $\psi(\langle w^*, x \rangle)$, in expectation over $w^*$. But, we know that $\psi(\langle w^*, x \rangle)$ can be written as a linear combination of ReLU neurons, so there must be some ReLU neuron which will be nearly uncorrelated with any linear combination of the random features (and as a result, cannot be well-approximated by them). Finally, by a symmetry argument, we can actually fix $w^*$ arbitrarily and the result still holds. We now turn to provide the formal proof:

*Proof of Theorem 4.2.* Take $\psi(x)$ from Proposition 4.6 and denote for $w \in \mathbb{R}^d$, $\psi_w(x) = \psi(\langle w, x \rangle)$ : $\mathbb{R}^d \to \mathbb{R}$. If we sample $w^*$ uniformly from $\{w : \|w\| = d\}$, then for all $f \in \mathcal{F}$:

$$\mathbb{E}_{w^*} [|\langle f_W, \psi_{w^*} \rangle|] \leq 20\|f_W\|^2 \exp(-c'd) \leq \exp(-cd),$$

where $c$ is a universal constant that depends only on the constant $c'$ from Proposition 4.6. Hence also:

$$\mathbb{E}_{w^*} [\mathbb{E}_W [|\langle f_W, \psi_{w^*} \rangle|]] \leq \exp(-cd)$$

We now show that $\mathbb{E}_W [|\langle f_W, \psi_{w^*} \rangle|]$ doesn't depend on $w^*$. Fix $w^*$, then any $w \in \mathbb{R}^d$ with $\|w\| = d$ can be written as $Mw^*$ for some orthogonal matrix $M$. Now:

$$\mathbb{E}_W [|\langle f_W, \psi_{w^*} \rangle|] \ \mathbb{E}_W [\mathbb{E}_x [|f(Wx) \cdot \psi(\langle w^*, x \rangle)|]] \ = \ \mathbb{E}_W [\mathbb{E}_x [|f(WM^TMx) \cdot \psi(\langle Mw^*, Mx \rangle)|]]$$
$$= \ \mathbb{E}_W [\mathbb{E}_x [|f(Wx) \cdot \psi(\langle Mw^*, x \rangle)|]] \ = \ \mathbb{E}_W [|\langle f_W, \psi_{Mw^*} \rangle|]$$

where we used the fact that both $x$ and the rows of $W$ have a spherically symmetric distribution. Therefore, for all $w^*$ with $\|w^*\| = d$:

$$\mathbb{E}_W [|\langle f_W, \psi_{w^*} \rangle|] \leq \exp(-cd) \tag{4}$$

Using Markov's inequality and dividing $c$ by a factor of 2, we get w.p $> 1 - \exp(-cd)$ over sampling of $W$, $|\langle f_W, \psi_{w^*}\rangle| \leq \exp(-cd)$. Finally, if we pick $f_1, \ldots, f_r \in \mathcal{F}$, using the union bound we get that w.p $> 1 - r\exp(-cd)$ over sampling of $W$:

$$\forall i \in \{1, \ldots, r\}, \ |\langle f_{i_W}, \psi_{w^*}\rangle| \leq \exp(-cd)$$

We can write $\psi(x) = \sum_{j=1}^{a} a_j[x + c_j]_+ - 1$, where $a = 6d^2 + 1$ and $|a_j| \leq 2$, $c_j \in [-a, a]$ for $j = 1, \ldots a$. Let $w^* \in \mathbb{R}^d$ with $\|w^*\| = d$, and denote $f_j^*(x) = [\langle w^*, x\rangle + c_j]_+$. Assume that for every $j$ we can find $u^j \in \mathbb{R}^r$ and $f_1, \ldots, f_r \in \mathcal{F}$ such that $\left\|\sum_{i=1}^{r} u_i^j f_{i_W} - f_j^*\right\|^2 \leq \epsilon$, where $W$ is distributed as above, otherwise there is a ReLU neuron that cannot be represented by a linear combination of random features, which is what we want to prove.. Let $f_0(Wx) = b_0$ the bias term of the output layer of the network, then also:

$$\mathbb{E}_x\left[\left(\sum_{i=0}^{r}\left(\sum_{j=1}^{a} u_i^j a_j\right) f_i(Wx) - \sum_{j=1}^{a} a_j f_j^*(x) - 1\right)^2\right] = \mathbb{E}_x\left[\left(\sum_{i=0}^{r} \widetilde{u}_i f_i(Wx) - \psi(\langle w^*, x\rangle)\right)^2\right]$$

$$\leq \epsilon \sum_{j=1}^{a} |a_j|^2 \leq 24d^2\epsilon \tag{5}$$

where $\widetilde{u}_i = \left(\sum_{j=1}^{a} u_i^j a_j\right)$. On the other hand using Eq. (4) and item 2 from Proposition 4.6 we get w.p $> 1 - r\exp(-cd)$ over the distribution of $W$ that:

$$\mathbb{E}_x\left[\left(\sum_{i=0}^{r} \widetilde{u}_i f_i(Wx) - \psi(\langle w^*, x\rangle)\right)^2\right] \geq \|\psi_{w^*}\|^2 - 2\left|\left\langle\sum_{i=0}^{r} \widetilde{u}_i f_{i_W}, \psi_{w^*}\right\rangle\right|$$

$$\geq d - 2\max_i|\widetilde{u}_i|\sum_{i=1}^{r}|\langle f_{i_W}, \psi_{w^*}\rangle| \geq d - 2\max_i|\widetilde{u}_i|r\exp(-cd)$$

$$\geq d - 4a\max_i|u_i^j|r\exp(-cd), \tag{6}$$

where the last inequality is true for all $j = 1, \ldots, a$. Combining Eq. (5) and Eq. (6), for all $w^* \in \mathbb{R}^d$ with $\|w^*\| = d$ there is $b^* \in [-a, a]$ such that w.p $> 1 - r\exp(-cd)$ over the sampling of $W$, if there is $u \in \mathbb{R}^r$ that:

$$\mathbb{E}_x\left[\left(\sum_{i=1}^{r} u_i f_i(Wx) - [\langle w^*, x\rangle + b^*]_+\right)^2\right] \leq \epsilon \tag{7}$$

then $r\max_i|u_i| \geq \left(1 - 24d\epsilon\right)\frac{1}{24d}\exp(cd)$.

Lastly, by multiplying both sides of Eq. (7) by $d$, using the homogeneity of ReLU and setting $\epsilon = \frac{1}{50d}$ we get that for every $\hat{w}^* \in \mathbb{R}^d$ with $\|\hat{w}^*\| = d^2$ there is $\hat{b}^* \in [-6d^3 - 1, 6d^3 + 1]$ such that w.p $> 1 - r\exp(-cd)$ over the sampling of $W$, if there is $\hat{u} \in \mathbb{R}^r$ such that

$$\mathbb{E}_x\left[\left(\sum_{i=1}^{r} \hat{u}_i f_i(Wx) - [\langle \hat{w}^*, x\rangle + \hat{b}^*]_+\right)^2\right] \leq \frac{1}{50},$$

then $r\max_i|\hat{u}_i| \geq \frac{1}{48d^2}\exp(cd)$. $\qquad\square$

**Remark 4.7.** *It is possible to trade-off between the norm of $w^*$ and the required error. This can be done by altering the proof of Theorem 4.2, where instead of multiplying both sides of Eq. (7) by $d$ we could have multiplied both sides by $\frac{\alpha}{d}$. This way the following is proved: With the same assumptions as in Theorem 4.2, for all $w^* \in \mathbb{R}^d$ with $\|w^*\| = \alpha$ there is $b^* \in \mathbb{R}$ with $|b^*| \leq 6\alpha d + 1$ such that for all $\epsilon \in \left(0, \frac{\alpha}{5d^2}\right)$ and all $f_1, \ldots, f_r \in \mathcal{F}$, w.h.p over sampling of $W$, if $\mathbb{E}_x\left[\left(\sum_{i=1}^{r} u_i f_i(Wx) - [\langle w^*, x\rangle + b^*]_+\right)^2\right] \leq \epsilon$, then $r \cdot \max_i|u_i| \geq (1 - \frac{5d^2}{\alpha}\epsilon)\frac{1}{8\alpha}\exp(c_3 d)$ for a universal constant $c_3$.*

## 4.3 General Features

In the previous subsection, we assumed that our features have a structure of the form $x \mapsto f(Wx)$ for a random matrix $W$. We now turn to a more general case, where we are given features of any kind without any assumptions on their internal structure, as long as they are sampled from some fixed distribution. We show that even in this setting, such features cannot capture single ReLU neurons in the worst-case (at the cost of proving this for some target weight vector $w^*$, instead of any $w^*$).

**Theorem 4.8.** *There exists a universal constant $c$ such that the following holds. Let $d > 40$, and let $\mathcal{F}$ be a family of functions from $\mathbb{R}^d$ to $\mathbb{R}$, such that $\|f\| \leq \exp(d/40)$ for all $f \in \mathcal{F}$. Also, for some $r \in \mathbb{N}$, let $D$ be an arbitrary distribution over tuples $(f_1, \ldots f_r)$ of functions from $\mathcal{F}$. Then there exists $w^* \in \mathbb{R}^d$ with $\|w^*\| = d^2$, and $b^* \in \mathbb{R}$ with $|b^*| \leq 6d^3 + 1$, such that with probability at least $1 - r \exp(-cd)$ over sampling $f_1, \ldots, f_r$, if*

$$\mathbb{E}_{x \sim \mathcal{N}(0,I)}\left[\left(\sum_{i=1}^{r} u_i f_i(x) - [\langle w^*, x \rangle + b^*]_+\right)^2\right] \leq \frac{1}{50} \ ,$$

*then*

$$r \cdot \max_i |u_i| \geq \frac{1}{48d^2}\exp\left(cd\right).$$

The proof is similar to the proof of Theorem 4.2. The main difference is that we do not have any assumptions on the distribution of the random features (as the assumption on the distribution on $W$ in Theorem 4.2), hence we can only show that there exists *some* ReLU neuron that cannot be well-approximated. On the other hand, we have almost no restrictions on the random features, e.g. they can be multi-layered neural network of any architecture and with any random initialization.

**Example 4.9.** *Besides generalizing the setting of the previous subsection, Theorem 4.8 also captures the setting of "linearized" neural tangent kernel (see Jacot et al. [20] and Subsection 2.1), where we consider a linear combination of functions of the form:*

$$\sigma'(\langle w_i, x \rangle)\langle a_i, x \rangle, \tag{8}$$

*where $w_i$ are randomaly chosen and $a_i$ are being optimized using gradient based methods. This is because each such function is a weighted sum of the features $f_{i,j}(x) = \sigma'(\langle w_i, x \rangle)x_j$ for $1 \leq j \leq d$ and $w_i$ are randomly initialized.*

### Acknowledgements

This research is supported in part by European Research Council (ERC) grant 754705. We thank Yuanzhi Li for some helpful comments on the previous version of this paper.

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
