[Supplementary Material · appendix.pdf]

# Appendices

## A   Comparison to Previous Works

### A.1   Optimization with Coupling

The method from Subsection 2.1 is used in several works, for example:

- Li and Liang [23] show a generalization bound for ReLU neural networks where there is a strong separability assumption on the distribution of the data, which is critical in the analysis.

- In Du et al. [15], it is shown that under some assumptions on the data, neural network with ReLU activation would reach a global minimum of the empirical risk in polynomial time.

- Allen-Zhu et al. [3] also show an empirical risk bound on a multi-layer feed-forward neural network with ReLU activation and relies on separability assumption on the data.

- Allen-Zhu et al. [2] show with almost no assumptions on the distribution of the data that the generalization of neural networks with two or three layers is better then the generalization of a "ground truth" function for a large family of functions, which includes polynomials.

- In Allen-Zhu and Li [1] similar techniques are used to show a generalization bound on recurrent neural networks.

- Cao and Gu [9] show a generalization result for multi-layered networks, where the generalization is compared to a family of functions that take an integral form, similar to the one developed in Theorem C.4. This integral form is also studied in [38], [21] in the context of studying neural networks as random features.

All the above fix the weights of the output layer and consider only the ReLU activation function. Note that while using ReLU as an activation function, it is possible to show that fixing the output layer does not change the expressive power of the network. With that said, in practice all layers of neural networks are being optimized, and the optimization process may not work as well if some of the layers are fixed.

### A.2   Optimization on all the Layers

The methods from Subsection 2.2 are also used in several works, for example:

In Andoni et al. [4] this approach is also used to prove that neural networks can approximate polynomials. There the weights are drawn from a complex distribution, thus optimization is done on a complex domain which is non standard. Moreover, that paper uses the exponent activation function and assumes that the distribution on the data is uniform on the complex unit circle.

In Daniely et al. [12] and Daniely [11] this approach is used to get a generalization bound with respect to a large family of functions, where the network may have more than two layers and a different architecture than simple feed-forward. The methods used there are through the "conjugate kernel", which is a function corresponding to the activation and architecture of the network. The proof of our result is relatively more direct, and gives a bound which correspond directly to the activation function, without going through the conjugate kernel. Moreover, those papers do not quantitatively characterize the class of polynomials learned by the network, with an explicit proof.

Du and Lee [14] rely on the same methods and assumptions as Du et al. [15] to show an empirical risk bound on multi-layer neural network where a large family of activations is considered and with several architectures, including ResNets and convolutional ResNets. However, this does not imply a bound on the population risk.

# B Proofs from section 4

*Proof of Proposition 4.1.* By definition of our norm and the fact that $\mathbb{E}[xx^\top] = I$ for a standard Gaussian distribution, it is enough to show that

$$\min_{u_1 \ldots u_r} \| \sum_{i=1}^{r} u_i w_i - w^* \|^2 \geq \frac{1}{4}.$$

Re-writing $\sum_{i=1}^{r} u_i w_i$ as $Wu$ (where $W$ is a $d \times r$ matrix and $u$ is a vector in $\mathbb{R}^d$), the left hand side is equivalent to $\min_u \|Wu - w^*\|^2$, which is a standard linear least square problem, with a minimum at $u = (W^\top W)^\dagger W^\top w^*$, and a minimal value of $\|(W(W^\top W)^\dagger W^\top - I)w^*\|^2$. Letting $W = USV^\top$ be an SVD decomposition of $W$ (where $U, V$ are orthogonal matrices) and simplifying a bit, we get:

$$\|(W(W^\top W)^\dagger W^\top - I)w^*\|^2 = \|(USV^\top(VSU^\top USV^\top)^\dagger VSU^\top - UU^\top)w^*\|^2$$
$$= \|U\|^2 \cdot \|(SV^\top(VS^2V^\top)^\dagger VSU^\top - U^\top)w^*\|^2 = \|MU^\top w^*\|^2,$$

where $M$ is (with probability 1) a fixed $d \times d$ diagonal matrix, with a diagonal composed of $d - r$ ones and $r$ zeros. Moreover, by symmetry, $U^\top w^*$ is uniformly distributed on the unit sphere, so it is enough to understand the distribution of $\|Mz\|^2$ for a random unit vector $z$. Since the function $z \mapsto \|Mz\|^2$ is 1-Lipschitz on the unit sphere, it follows by standard concentration results for Lipschitz functions (see for example [22]) that with probability at least $1 - \exp(-cd)$ (for some universal constant $c$), $\mathbb{E}[\|Mz\|^2] - \|Mz\|^2 \leq \frac{1}{4}$. Finally, $\mathbb{E}[\|Mz\|^2] \geq \frac{1}{2}$, since clearly $\mathbb{E}[\|Mz\|^2] \geq \mathbb{E}[\|(I - M)z\|^2]$ (note that $M$ has more ones than zeros on the diagonal, as $d - r \geq r$), yet $\mathbb{E}[\|Mz\|^2] + \mathbb{E}[\|(I - M)z^2\|] = \mathbb{E}[\|z\|^2] = 1$. Combining the above, the result follows. $\qquad\square$

In the proof of Proposition 4.6 we rely on the following claim from [33, Lemma 5] [1]:

**Claim B.1.** *For any $f \in L^2(\mathbb{R}^d)$, and odd periodic function $\psi : \mathbb{R} \to \mathbb{R}$ if $d > 40$, and we sample $w\mathbb{R}^d$ uniformly from $\{w : \|w\| = 2r\}$, it holds that:*

$$\mathbb{E}_w\left(\langle f(\cdot), \psi(\langle w, \cdot\rangle)\rangle^2\right) \leq 10\|f\|^2 \cdot \left(\exp(-d/20) + \sum_{n=1}^{\infty} \exp(-nr^2)\right)$$

*for a universal constant $c > 0$.*

*Proof of Proposition 4.6.* For every $x_0 \in [-a, a - 4]$ we have that:

$$\psi(x_0 + 4) = [x_0 + 4 + a]_+ + \sum_{n=1}^{a} 2[x_0 + 4 + a - 2n]_+(-1)^n - 1$$

$$= x_0 + a + 4 + \sum_{n=1}^{\lfloor \frac{x_0+a}{2} \rfloor + 2} 2(x_0 + 4 + a - 2n)(-1)^n - 1$$

$$= x_0 + a + \sum_{n=1}^{\lfloor \frac{x_0+a}{2} \rfloor} 2(x_0 + a - 2n) - 1 + 4+$$

$$+ \sum_{n=1}^{\lfloor \frac{x_0+a}{2} \rfloor} 8(-1)^n + (-1)^{\lfloor \frac{x_0+a}{2} \rfloor + 1} 2(x_0 + 4 + a - (x_0 + a + 2))+$$

$$+ (-1)^{\lfloor \frac{x_0+a}{2} \rfloor + 2} 2(x_0 + 4 + a - (x_0 + a + 4))$$

$$= [x_0 + a]_+ + \sum_{n=1}^{a} 2[x_0 + a - 2n]_+ - 1 = \psi(x_0),$$

where we used the fact that $a$ is odd. This proves that $\psi(x)$ is periodic in $[-a, a]$ with a period of $4$. To prove that $\psi(x)$ is an odd function, note that:

$$\psi(-2) = \psi(2) = \psi(0) = 0$$

and also that $\psi(1) = -\psi(-1) = \pm 1$, and between every two integers $\psi(x)$ is a linear function. The exact value of $\psi(1)$ and $\psi(-1)$ depends on whether $\frac{a+1}{2}$ is even or odd. Thus, $\psi(x)$ is an odd function in the interval $[-2, 2]$, and because it is periodic with a period of $4$, it is also an odd function in the interval $[-a, a]$. This proves item 1 in the proposition.

For item 2, using the fact that $x \in \mathbb{R}^d$ is distributed symmetrically we can assume w.l.o.g that

$w^* = \begin{pmatrix} d \\ 0 \\ \vdots \\ 0 \end{pmatrix}$. Thus,

$$
\begin{aligned}
\|\psi(\langle w^*, \cdot \rangle)\|^2 &= c_d \int_{x \in \mathbb{R}^d} |\psi(\langle w^*, x \rangle)|^2 e^{\frac{-\|x\|^2}{2}} dx \\
&= c_d \int_{-\infty}^{\infty} |\psi(dx_1)|^2 e^{\frac{-x_1^2}{2}} dx_1 \cdot \int_{\infty}^{\infty} e^{\frac{-x_2^2}{2}} dx_2 \cdots \int_{\infty}^{\infty} e^{\frac{-x_d^2}{2}} dx_d \\
&\geq \frac{1}{d\sqrt{2\pi}} \int_{-\infty}^{\infty} |\psi(x_1)|^2 e^{-\left(\frac{x_1}{d}\right)^2} dx_1 \geq \frac{1}{d\sqrt{2\pi}} \int_{-a}^{a} |\psi(x_1)|^2 e^{-1} dx_1 \\
&\geq \frac{1}{d\sqrt{2\pi}} a e^{-1} \cdot \frac{4}{3} \geq \frac{1}{d\sqrt{2\pi}} 6d^2 e^{-1} \cdot \frac{4}{3} \geq d.
\end{aligned}
$$

In the above, we used the fact that for every interval of the form $[n, n+2]$ for $n \in [-a, a-2]$, the integral

$$\int_{[n,n+2]} |\psi(x_1)|^2 dx_1 = \frac{4}{3},$$

and there are $a$ such intervals in $[-a, a]$.

For item 3, define $\widetilde{\psi}(x)$ to be equal to $\psi(x)$ on $[-a, a]$, and continue it to $[-\infty, \infty]$ such that it is periodic. Let $g(x) = \widetilde{\psi}(x) - \psi(x)$, then $g(x) = 0$ for $x \in [-a, a]$ and $|g(x)| \leq x$ for $|x| > a$. Now for every $w \in \mathbb{R}^d$ with $\|w\| = d$ we get that:

$$
\begin{aligned}
\|g(\langle w, \cdot \rangle)\|^2 = \mathbb{E}_x \left[ g^2(\langle w, x \rangle) \right] &\leq \mathbb{E}_x \left[ \mathbb{1}_{|\langle w, x \rangle| \geq a} \langle w, x \rangle^2 \right] \\
&\leq \sqrt{\mathbb{E}_x \left[ \mathbb{1}^2_{|\langle w, x \rangle| \geq a} \right]} \cdot \sqrt{\mathbb{E}_x \left[ \langle w, x \rangle^4 \right]} \\
&\leq \sqrt{\mathbb{E}_x \left[ \mathbb{1}_{|\langle w, x \rangle| \geq a} \right]} \cdot \|w\|^2 \sqrt{\mathbb{E}_x \left[ \langle \frac{w}{\|w\|}, x \rangle^4 \right]} \qquad (9) \\
&= 3d^2 \sqrt{P\left( |\langle w, x \rangle| \geq a \right)} \qquad (10)
\end{aligned}
$$

where we used the fact that since $x \sim N(0, I_d)$ then $\langle \frac{w}{\|w\|}, x \rangle$ has a standard Gaussian distribution, hence its fourth moment is 3. Also $\langle w, x \rangle \sim N(0, d)$, Hence

$$P\left( |\langle w, x \rangle| \geq 6d^2 + 1 \right) \leq \exp(-d),$$

and using Eq. (10) we get:

$$\|g(\langle w, \cdot \rangle)\|^2 \leq 3d^2 \exp(-d) \leq \exp(-cd), \qquad (11)$$

for a constant $c$. Now for every $f \in L^2(\mathbb{R}^d)$:

$$\mathbb{E}_w \left( \langle f(\cdot), \psi(\langle w, \cdot \rangle) \rangle^2 \right) \leq 2\mathbb{E}_w \left( \langle f(\cdot), \widetilde{\psi}(\langle w, \cdot \rangle) \rangle^2 \right) + 2\mathbb{E}_w \left( \langle f(\cdot), \widetilde{\psi}(\langle w, \cdot \rangle) - \psi(\langle w, \cdot \rangle) \rangle^2 \right). \qquad (12)$$

Using Cauchy-Schwartz and Eq. (11) we can bound the second term:

$$\mathbb{E}_w \left( \langle f(\cdot), \widetilde{\psi}(\langle w, \cdot \rangle) - \psi(\langle w, \cdot \rangle) \rangle^2 \right) = \mathbb{E}_w \left( \langle f(\cdot), g(\langle w, \cdot \rangle)^2 \right) \leq \|f\|^2 \exp(-cd),$$

and finally using Claim B.1 on $\widetilde{\psi}(x)$, and taking $r = d$ we can bound the first term of Eq. (12), by changing the constant from the claim by a factor of at most 4. Thus, there exists a universal constant $c$ such that:

$$\mathbb{E}_w \left( \langle f(\cdot), \psi(\langle w, \cdot \rangle) \rangle^2 \right) \leq \|f\|^2 \exp(-cd)$$

$\square$

*Proof of Theorem 4.8.* Take $\psi(x)$ from Proposition 4.6 and denote for $w \in \mathbb{R}^d$, $\psi_w(x) = \psi(\langle w, x \rangle)$. If we sample $w^*$ uniformly from $\{w : \|w\| = d\}$ and $(f_1, \dots, f_r) \sim D$ then:

$$\mathbb{E}_{w^*} \left[ \mathbb{E}_{(f_1,\dots,f_r)} \left[ \left| \langle \sum_{i=1}^{r} f_i, \psi_{w^*} \rangle \right| \right] \right] = \mathbb{E}_{(f_1,\dots,f_r)} \left[ \mathbb{E}_{w^*} \left[ \left| \langle \sum_{i=1}^{r} f_i, \psi_{w^*} \rangle \right| \right] \right]$$

$$\leq \mathbb{E}_{(f_1,\dots,f_r)} \left[ 20 \left\| \sum_{i=1}^{r} f_i \right\| \exp(-c'd) \right] \leq \mathbb{E}_{(f_1,\dots,f_r)} \left[ \sum_{i=1}^{r} 20 \|f_i\| \exp(-c'd) \right] \leq r \exp(-cd)$$

where $c$ is a universal constant that depends only on the constant $c'$ from Proposition 4.6. Thus, there exists $w^*$ such that:

$$\mathbb{E}_{(f_1,\dots,f_r)} \left[ \left| \langle \sum_{i=1}^{r} f_i, \psi_{w^*} \rangle \right| \right] \leq r \exp(-cd. \tag{13}$$

Using Markov's inequality on Eq. (13), with the fixed $w^*$ that was found and dividing $c$ by a factor of 2, we get w.p $> 1 - r \exp(-cd)$ over sampling of $(f_1, \dots, f_r) \sim D$ that:

$$\left| \langle \sum_{i=1}^{r} f_i, \psi_{w^*} \rangle \right| \leq r \exp(-cd)$$

The rest of the proof is the same as the proof of Theorem 4.2, except for fixing the $w^*$ we found above. $\square$

## C  Neural Networks Learn Polynomials

The data for our network is $(x, y) \in \mathbb{R}^d \times \mathbb{R}$, drawn from an unknown distribution $D$. We assume for simplicity that $\|x\| \leq 1$ and $y = \{-1, +1\}$.

We consider one-hidden-layer feed-forward neural networks which are defined as:

$$N(x) = N(W, U, x) = U\sigma(Wx),$$

where $\sigma$ is an activation function which acts coordinate-wise and $W \in \mathbb{R}^{r \times d}$, $U \in \mathbb{R}^r$. We will also use the following form for the network:

$$N(x) = \sum_{i=1}^{r} u_i \sigma(\langle w_i, x \rangle) \tag{14}$$

here $u_i \in \mathbb{R}$ and $w_i \in \mathbb{R}^d$.

For simplicity we will use the hinge loss, which is defined by: $l(\hat{y}, y) = \max\{0, 1 - \hat{y}y\}$, thus the optimization will be done on the function $l(N(x), y) = l(N(W, U, x), y)$. We will also use the notation:

$$L_D(W, U) = \mathbb{E}_{(x,y) \sim D} [l(N(W, U, x), y)]$$

We will use the standard form of SGD to optimize $L_D$, where at each iteration a random sample $(x_i, y_i)$ is drawn from $D$ and we update:

$$W_{i+1} = W_i - \eta \frac{\partial l(N(W_i, U_i, x_i), y_i)}{\partial W_i}$$

$$U_{i+1} = U_i - \eta \frac{\partial l(N(W_i, U_i, x_i), y_i)}{\partial U_i}$$

The initialization of $W_0$ is a standard Xavier initialization [18], that is $w_i \sim U\left(\left[\frac{-1}{\sqrt{d}}, \frac{1}{\sqrt{d}}\right]^d\right)$. $U_0$ can be initialized in any manner, as long as its norm is smaller than $\frac{1}{\sqrt{r}}$, e.g. we can initialize $U_0 = 0$. This kind of initialization for the outer layer has been used also in other works (see [11], [4]).

The main result of this section is the following:

**Theorem C.1** (Formal statement of Theorem 3.1). *Let $\sigma : \mathbb{R} \to \mathbb{R}$ be an analytic activation function, which is $L$-Lipschitz with $\sigma(0) \leq L$. Let $D$ be any distribution over the labelled data $(x,y) \in \mathbb{R}^d \times \mathbb{R}$ with $\|x\| \leq 1$, $y \in \{-1, +1\}$, and let $\epsilon > 0$, $\delta > 0$, $\alpha > 0$, and $k$ be some positive integer. Suppose we run SGD on the neural network:*

$$N(W, U, x) = U\sigma(Wx) = \sum_{i=1}^{r} u_i \sigma(\langle w_i, x \rangle)$$

*with the following parameters:*

1. *$r$ neurons with $r \geq \frac{64\beta^6 L^2}{\epsilon^4} \log\left(\frac{1}{\delta}\right)$*

2. *$W_0$ is initialized with $w_i \sim U\left(\left[\frac{-1}{\sqrt{d}}, \frac{1}{\sqrt{d}}\right]^d\right)$ for $i = 1, \ldots, r$ and $U_0$ is initialized s.t*
   *$\|U_0\| \leq \frac{1}{\sqrt{r}}$*

3. *learning rate $\eta = \frac{\epsilon}{8r}$*

4. *$T$ steps with $T = \frac{4\beta^2}{\epsilon^2}$*

*here $\beta = \alpha^k \left(\frac{A}{a}\right)^k (12d)^{2k^2}$, where $a \leq a_i \leq A$, bounds the first $k$ coefficients $a_1, \ldots, a_k$ of the Taylor series expansion of the activation $\sigma$. Then for every polynomial $P(x_1, \ldots, x_d)$ with $\deg(P) \leq k$, the coefficients of $P$ are bounded by $\alpha$ and all the monomials of $P$ which have a non-zero coefficient also have a non-zero coefficient in the Taylor series of $\sigma$, w.p $> 1 - \delta$ over the initialization there is $t \in [T]$ such that:*

$$\mathbb{E}\left[L_D\left(W_t, U_t\right)\right] \leq L_D(P(x)) + \epsilon.$$

*Here the expectation is over the random choice of $(x_i, y_i)$ in each iteration of SGD.*

We note that for simplicity, we focused on analytic activation functions, although it is possible to derive related results for non-analytic activations such as a ReLU (see Appendix G for a discussion). Also, note that we did not use a bias term in the architecture of the network in the theorem (namely, we have $\sigma(\langle w_i, x \rangle)$ and not $\sigma(\langle w_i, x \rangle + b_i)$). This is because if the polynomial we are trying to compete with has a constant factor, then we require that the Taylor expansion of the activation also has a constant factor, thus the bias term is already included in the Taylor expansion of the activation function.

**Remark C.2.** *Suppose we are given a sample set $S = \{(x_i, y_i)\}_{i=1}^{m}$. By choosing $D$ uniform on the sample set $S$, Theorem C.1 shows that SGD over the sample set will lead to an average loss not much worse than the best possible polynomial predictor with bounded degree and coefficients.*

At high level, the proof idea of Theorem C.1 is divided into three steps. In the first step we show that with an appropriate learning rate and limited amount of iterations, neural networks generalize better than random features. This step allows us to focus our analysis on the behaviour of a linear combination of random features instead of the more complicated architecture of neural networks. In the second step using McDiarmid's theorem we show that by taking enough random features, they concentrate around their expectation. In the third step we use Legendre's polynomials to show that any polynomial can be represented as an expectation of random features. For a full proof see Appendix C.

First we introduce some notations regarding multi-variable polynomials: Letting $J = (j_1, \ldots, j_d)$ be a multi index, and given $x \in \mathbb{R}^d$, we define $x^J = x_1^{j_1} \cdots x_d^{j_d}$, and also $|J| = j_1 + \cdots + j_d$. We say for two multi indices $J', J$ that $J' \leq J$ if for all $1 \leq i \leq d$, $j_i' \leq j_i$ and that $J' < J$ if $J' \leq J$ and also there is an index $1 \leq s \leq d$ such that $j_s' < j_s$. For $k \in \mathbb{N}$ and multi index $J = (j_1, \ldots, j_d)$

we say that $J \leq k$ if $j_1 + \ldots j_d \leq k$. Lastly, given a multi-variable polynomials $P(x) = \sum_J c_J x^J$, where $c_J \in \mathbb{R}$ we define:
$$|P| = \max_J |c_J|.$$

We break the proof to three steps, where each step contains a theorem which is independent of the other steps. Finally we combine the three steps to prove the main theorem.

**Step 1: SGD on Over-Parameterized Networks Competes with Random Features**

Recall we use a network of the form:

$$N(W, U, x) = U\sigma(Wx) = \sum_{i=1}^r u_i \sigma(\langle w_i, x \rangle), \tag{15}$$

where $W_0$, $U_0$ are initialized as described in the theorem We show that for any target matrix $U^*$ with a small enough norm and every $\epsilon > 0$, if we run SGD on $l(N(W_0, U_0, x), y)$ with appropriate learning rate $\eta$ and number of iterations $T$, there is some $t \in [T]$ with:

$$\mathbb{E}\left(L_D(W_t, U_t)\right) \leq L_D(W_0, U^*) + \epsilon \tag{16}$$

where the expectation is over the random choices of examples in each round of SGD.

The bound in Eq. (16) means that SGD on randomly initialized weights competes with random features. By random features here we mean any linear combination of neurons of the form $\sigma(\langle w_i, x \rangle)$ where the $w_i$ are randomly chosen, and the norm of the weights of the linear combination are bounded. In more details:

**Theorem C.3.** *Assume we initialize $U_0, W_0$ such that $\|U_0\| \leq \frac{1}{\sqrt{r}}$ and $\|W_0\| \leq \sqrt{r}$. Also assume that $\sigma$ is $L$-Lipschitz with $\sigma(0) \leq L$, and let $C \geq 1$ be a constant. Letting $\epsilon > 0$, we run SGD with step size $\eta = \frac{\epsilon}{8r}$ and $T$ steps with $T = \frac{4C^2}{\epsilon^2}$ and let $W_1, \ldots, W_T$ be the weights produced at each step. If we pick $r$ such that $r \geq \frac{64C^6 L^2}{\epsilon^4}$ then for every target matrix $U^*$ with $\|U^*\| \leq \frac{C}{\sqrt{r}}$ there is a $t \in [T]$ s.t:*
$$\mathbb{E}\left[L_D(W_t, U_t)\right] \leq L_D(W_0, U^*) + \epsilon.$$

*Here the expectation is over the random choice of the training examples in each round of SGD.*

In the proof of Theorem C.3 we first show that for the chosen learning rate $\eta$ and limited number of iterations $T$, the matrix $W$ does not change much from its initialization. After that we use results from online convex optimization for linear prediction with respect to $U^*$ with a sufficiently small norm to prove the required bound. For a full proof see Appendix F. Note that in the theorem we did not need to specify the initialization scheme, only to bound the norm of the initialized weights. The optimization analysis is similar to the one done in Daniely [11].

**Step 2: Random Features Concentrate Around their Expectation**

In the previous step we showed that in order to bound the expected loss of the network, it is enough to consider a network of the form $\sum_i u_i \sigma(\langle w_i, x \rangle)$, where the $w_i$ are randomly initialized with $w_i \sim U\left(\left[\frac{-1}{\sqrt{d}}, \frac{1}{\sqrt{d}}\right]^d\right)$. We now show that if the number of random features $r$ is large enough, then a linear combination of them approximates functions of the form $x \mapsto \mathbb{E}_w[\sigma(\langle w, x \rangle)g(w)] = c_d \int_{w \in \left[\frac{-1}{\sqrt{d}}, \frac{1}{\sqrt{d}}\right]^d} g(w)\sigma(\langle w, x \rangle)dw$ for an appropriate normalization factor $c_d$:

**Theorem C.4.** *Let $f(x) = c_d \int_{w \in \left[\frac{-1}{\sqrt{d}}, \frac{1}{\sqrt{d}}\right]^d} g(w)\sigma(\langle w, x \rangle)dw$ where $\sigma : \mathbb{R} \to \mathbb{R}$ is $L$-Lipschitz on $[-1, 1]$ with $\sigma(0) \leq L$, and $c_d = \left(\frac{\sqrt{d}}{2}\right)^d$ a normalization term. Assume that $max_{\|w\| \leq 1} |g(w)| \leq C$ for a constant $C$. Then for every $\delta > 0$ if $w_1, \ldots, w_r$ are drawn i.i.d from the uniform distribution on $\left[\frac{-1}{\sqrt{d}}, \frac{1}{\sqrt{d}}\right]^d$, w.p $> 1 - \delta$ there is a function of the form*

$$\hat{f}(x) = \sum_{i=1}^r u_i \sigma(\langle w_i, x \rangle)$$

*where $|u_i| \leq \frac{C}{r}$ for every $1 \leq i \leq r$, such that:*

$$\sup_x \left| \hat{f}(x) - f(x) \right| \leq \frac{LC}{\sqrt{r}} \left( 4 + \sqrt{2 \log \left( \frac{1}{\delta} \right)} \right)$$

Theorem C.4 basically states that random features concentrate around their expectation, and the rate of convergence is $O\left( \frac{1}{\sqrt{r}} \right)$ where $r$ is the amount of random features that were sampled. The proof is based on concentration of measure and Rademacher complexity arguments, and appears in Appendix E.

## Step 3: Representing Polynomials as Expectation of Random Features

In the previous step we showed that random features can approximate functions with the integral form:

$$f(x) = c_d \int_{w \in \left[ \frac{-1}{\sqrt{d}}, \frac{1}{\sqrt{d}} \right]^d} g(w) \sigma(\langle w, x \rangle) dw$$

In this step we show how a a polynomial $P(x)$ with bounded degree and coefficients can be represented in this form. This means that we need to find a function $g(w)$ for which $f(x) = P(x)$. To do so we use the fact that $\sigma(x)$ is analytic, thus it can be represented as an infinite sum of monomials using a Taylor expansion, and take $g(w)$ to be a finite weighted sum of Legendre polynomials, which are orthogonal with respect to the appropriate inner product. The main difficulty here is to find a bound on $\max_{w \in \left[ \frac{-1}{\sqrt{d}}, \frac{1}{\sqrt{d}} \right]^d} |g(w)|$, which in turn also bounds the distance between the sum of the random features and its expectation. The main theorem of this step is:

**Theorem C.5.** *Let $\sigma : \mathbb{R} \to \mathbb{R}$ be an analytic function, and $P(x) = \sum_J \alpha_J x^J$ be a polynomial, where $x \in \mathbb{R}^d$, and all the monomials of $P$ which have a non-zero coefficient also have a non-zero coefficient in the Taylor series of $\sigma$. Assume that $\deg(P) \leq k$ and $|\alpha_J| \leq \alpha$ for all J, with $\alpha \geq 1$. Then there exists a function $g(w) : \mathbb{R}^d \to \mathbb{R}$ that satisfies the following:*

1. $c_d \int_{w \in \left[ \frac{-1}{\sqrt{d}}, \frac{1}{\sqrt{d}} \right]^d} \sigma(\langle w, x \rangle) g(w) dw = P(x)$

2. $\max_{w \in \left[ \frac{-1}{\sqrt{d}}, \frac{1}{\sqrt{d}} \right]^d} |g(w)| \leq \alpha^k \left( \frac{A}{a} \right)^k (12d)^{2k^2}$

*Here $c_d = \left( \frac{\sqrt{d}}{2} \right)^d$ is a normalization term, and $a \leq a_i \leq A$, bounds the first k coefficients $a_1, \ldots, a_k$ of the Taylor series expansion of the activation $\sigma$.*

For a full proof of Theorem C.5 and an overview of Legendre polynomials see Appendix D.

## Step 4: Putting it all Together

We are now ready to prove the main theorem of this section. The proof is done for convenience in reverse order of the three steps presented above.

*Proof of Theorem C.1.* Let $a_0, a_1, \ldots, a_k$ be the coefficients of the Taylor expansion of $\sigma$ up to degree $k$, and let $P(x)$ be a a polynomial with $\deg(P) \leq k$ and $|P| \leq \alpha$, such that if $a_j = 0$ then the monomials in $P(x)$ of degree $j$ also have a zero coefficient.

First, we use Theorem C.5 to find a function $g(w)$ such that:

$$c_d \int_{w \in \left[ \frac{-1}{\sqrt{d}}, \frac{1}{\sqrt{d}} \right]^d} \sigma(\langle w, x \rangle) g(w) = P(x). \tag{17}$$

Then we consider drawing random features $w_1, \ldots, w_r \sim U\left(\left[\frac{-1}{\sqrt{d}}, \frac{1}{\sqrt{d}}\right]^d\right)$ i.i.d. Using Theorem C.4, the choice of $r$ and Eq. (17), w.p $> 1 - \delta$ there is $U^* = (u_1, \ldots, u_r)$ such that:

$$\sup_x \left| \sum_{i=1}^r u_i \sigma(\langle w_i, x \rangle) - P(x) \right|$$

$$= \sup_x \left| \sum_{i=1}^r u_i \sigma(\langle w_i, x \rangle) - c_d \int_{w \in \left[\frac{-1}{\sqrt{d}}, \frac{1}{\sqrt{d}}\right]^d} \sigma(\langle w, x \rangle) g(w) \right| \leq \epsilon, \qquad (18)$$

and also $|u_i| \leq \max_{\|w\| \leq 1} \frac{|g(w)|}{r} \leq \alpha^k \left(\frac{A}{a}\right)^k (12d)^{2k^2}$, thus $\|U^*\| \leq \frac{\alpha^k \left(\frac{A}{a}\right)^k (12d)^{2k^2}}{\sqrt{r}}$.

Finally, we use Theorem C.3 with the defined learning rate $\eta$ and iterations $T$ to find $t \in [T]$ such that:

$$\mathbb{E}\big(L_D(W_t, U_t)\big) \leq L_D(W_0, U^*) + \epsilon. \qquad (19)$$

Combining Eq. (18) with Eq. (19) gives:

$$\mathbb{E}\big(L_D(W_t, U_t)\big) \leq L_D(P(x)) + 2\epsilon.$$

Re-scaling $\epsilon$ finishes the proof. $\qquad\square$

## D   Representing Polynomials as Expectation of Random Features

We will use the Legendre polynomials in the one variable and multi-variable case. Let $p_1(w), \ldots, p_k(w)$ be the one variable Legendre polynomials, these polynomials are an orthogonal basis for one variable polynomials with respect to the inner product:

$$\langle f, g \rangle = \int_{-1}^1 f(w)g(w)dw.$$

They are normalized by $p_0(w) = 1$, and their inner product is:

$$\langle p_i, p_j \rangle = \delta_{i,j} \frac{2}{2i + 1},$$

where $\delta_{i,j} = 1$ if $i = j$ and 0 otherwise. Let $J = (j_1, \ldots, j_d)$ be a multi index, and define for $w \in \mathbb{R}^d$:

$$p_J(w) = p_{j_1}(w_1) \cdots p_{j_d}(w_d).$$

Using tensor product of polynomial spaces, the polynomials $p_J$ for all multi-indices $J$ form an orthogonal base for $d$-dimensional polynomials (see [16]), with respect to the inner product:

$$\langle f, g \rangle = \int_{w \in [-1,1]^d} f(w)g(w)dw.$$

The next lemma gives the coefficient of the Legendre expansion of monomials in one variable:

**Lemma D.1.** *Let:*

$$I(m, n) = \int_{-1}^1 w^m p_n(w)dw \qquad (20)$$

*Then:*

1. *If $m < n$ or $m + n$ is odd then $I(m, n) = 0$*

2. *If $m \geq n$ and $m + n$ is even then:*

$$I(m, n) = 2^{n+1}(n + 1) \frac{m! \left(\frac{m+n+1}{2}\right)!}{\left(\frac{m-n}{2}\right)!(m + n + 1)!}$$

*and in this case: $\frac{1}{2^m} \leq I(m, n) \leq 1$*

*Proof.* If $m < n$ then $w^m$ is in the span of $p_0(w), \ldots, p_n(w)$ thus by orthogonality of the Legendre polynomials we get that $I(m,n) = 0$.

If $m + n$ is odd, then $w^m p_n(w)$ is an odd polynomial, thus its integral over a symmetric interval is zero, this concludes the first case.

Assume now that $m \geq n$ and $m + n$ is even. Plugging Rodrigues formula:

$$p_n(w) = \frac{1}{2^n n!} \left( \frac{d^n}{dw^n} \left( w^2 - 1 \right)^n \right),$$

in Eq. (20) and doing integration by parts $n$ times yields:

$$I(m,n) = \frac{1}{2^n} \binom{m}{n} \int_{-1}^{1} w^{m-n} (w^2 - 1)^n dw.$$

By the assumption $m - n$ is even, we can do a change of variables $t = w^2$ to get a beta function integral:

$$I(m,n) = \frac{1}{2^n} \binom{m}{n} B \left( \frac{m-n+1}{2}, n+1 \right) = \frac{1}{2^n} \frac{m!}{n!(m-n)!} \frac{\left( \frac{m-n+1}{2} \right)!(n+1)!}{\left( \frac{m+n+3}{2} \right)!}. \tag{21}$$

Using the known identity about Gamma function

$$\left( n + \frac{1}{2} \right)! = \Gamma \left( n + \frac{1}{2} \right) = \frac{(2n)!}{4^n n!} \sqrt{\pi},$$

and plugging it into Eq. (21), we get:

$$I(m,n) = 2^{n+1}(n+1) \frac{m! \left( \frac{m+n+1}{2} \right)!}{\left( \frac{m-n}{2} \right)!(m+n+1)!}. \tag{22}$$

Note that for a given $m$, $I(m,n)$ is largest when $n = 0$ and smallest when $n = m$. Plugging $n = 0$ gives:

$$I(m,0) = \frac{m! \left( \frac{m+1}{2} \right)!}{(m+1)! \left( \frac{m}{2} \right)!} \leq 1.$$

Plugging $n = m$ and using the fact the $\binom{2m}{m} \leq 4^m$ gives:

$$I(m,m) = 2^{m+1}(m+1) \frac{m! \left( m + \frac{1}{2} \right)!}{(2m+1)!} \geq 2^m \frac{2m+2}{2m+1} \binom{2m}{m}^{-1} \geq \frac{1}{2^m}.$$

$\square$

*Proof of Theorem C.5.* Let $N \subset \mathbb{N}$ be all the indices smaller than $k$ for which the coefficient of the Taylor expansion $a_i$ of the activation $\sigma$ is non-zero. We define:

$$g(w) = \sum_{|J| \leq k} c_J p_J \left( \sqrt{d} w \right) \tag{23}$$

where $p_J(w)$ are the multi-variable Legendre polynomials, and the coefficients $c_J$ will be determined later. Since $\sigma$ is analytic we can use its Taylor expansion to get:

$$c_d \int_{w \in \left[ \frac{-1}{\sqrt{d}}, \frac{1}{\sqrt{d}} \right]^d} \sigma(\langle w, x \rangle) g(w) dw = c_d \int_{w \in \left[ \frac{-1}{\sqrt{d}}, \frac{1}{\sqrt{d}} \right]^d} \sum_{i=1}^{\infty} a_i \langle w, x \rangle^i g(w) dw$$

$$= c_d \sum_{i \in N} a_i \int_{w \in \left[ \frac{-1}{\sqrt{d}}, \frac{1}{\sqrt{d}} \right]^d} \langle w, x \rangle^i g(w) dw \tag{24}$$

Plugging $g(w)$ and $c_d$ into Eq. (24) and using the change of variables $\sqrt{d}w \mapsto w$ gives:

$$\left(\frac{\sqrt{d}}{2}\right)^d \sum_{i \in N} a_i \int_{w \in \left[\frac{-1}{\sqrt{d}}, \frac{1}{\sqrt{d}}\right]^d} \langle w, x \rangle^i g(w) dw$$

$$= \left(\frac{\sqrt{d}}{2}\right)^d \sum_{i \in N} a_i \int_{w \in \left[\frac{-1}{\sqrt{d}}, \frac{1}{\sqrt{d}}\right]^d} \langle w, x \rangle^i \sum_{|J'| \leq k} c_{J'} p_{J'} \left(\sqrt{d}w\right) dw$$

$$= \left(\frac{1}{2}\right)^d \sum_{i \in N} \frac{a_i}{\left(\sqrt{d}\right)^i} \int_{w \in [-1,1]^d} \langle w, x \rangle^i \sum_{|J'| \leq k} c_{J'} p_{J'}(w) dw$$

$$= \left(\frac{1}{2}\right)^d \sum_{|J|=i,\ i \in N} \frac{a_i}{\left(\sqrt{d}\right)^i} x^J \int_{[-1,1]^d} w^J \sum_{|J'| \leq k} c_{J'} p_{J'}(w) dw. \tag{25}$$

Now we use the Legendre expansion of monomials. Letting $J = (j_1, \ldots, j_d)$ be a multi-index, expand $w^J$ in the multi-variable Legendre basis:

$$w^J = \sum_{J' \leq J} b_{J,J'} p_{J'}(w). \tag{26}$$

Plugging this into Eq. (25) gives:

$$\left(\frac{\sqrt{d}}{2}\right)^d \sum_{i \in N} a_i \int_{w \in \left[\frac{-1}{\sqrt{d}}, \frac{1}{\sqrt{d}}\right]^d} \langle w, x \rangle^i g(w) dw$$

$$= \left(\frac{1}{2}\right)^d \sum_{|J|=i,\ i \in N} \frac{a_i}{\left(\sqrt{d}\right)^i} x^J \int_{[-1,1]^d} \sum_{J'' \leq J} b_{J,J''} p_{J''}(w) \sum_{|J'| \leq k} c_{J'} p_{J'}(w) dw$$

$$= \sum_{|J|=i,\ i \in N} x^J \sum_{J' \leq J} \left(\frac{1}{2}\right)^d \frac{a_i}{\left(\sqrt{d}\right)^i} c_{J'} b_{J,J'} \int_{w \in [-1,1]^d} p_{J'}^2(w) dw \tag{27}$$

where in the last equality we used the orthogonality of the Legendre polynomials.
Denote $\|p_J\|^2 = \int_{w \in [-1,1]^d} p_J^2(w) dw$. In order to show that item 1 holds (using Eq. (27)) we need to show that for every multi index $J$ that appears in $P(x)$ we can choose $c_{J'}$ for $J' \leq J$ such that:

$$\sum_{J' \leq J} \left(\frac{1}{2}\right)^d \frac{a_{|J|}}{\left(\sqrt{d}\right)^{|J|}} c_{J'} b_{J,J'} \|p_{J'}\|^2 = \alpha_J.$$

By induction on $J$, for $J = (0, \ldots, 0)$, take:

$$c_J = 2^d \alpha_J \cdot \left(a_0 b_{J,J} \|p_J\|^2\right)^{-1} \tag{28}$$

Assume we defined $c_{J'}$ for every $J' < J$, then we define:

$$c_J = \left(\alpha_J - \left(\frac{1}{2}\right)^d \sum_{J' < J} \frac{a_{|J|}}{\left(\sqrt{d}\right)^{|J|}} b_{J,J'} c_{J'} \|p_{J'}\|^2\right) \cdot \left(\left(\frac{1}{2}\right)^d \frac{a_{|J|}}{\left(\sqrt{d}\right)^{|J|}} b_{J,J} \|p_J\|^2\right)^{-1} \tag{29}$$

This shows that item 1 holds.

For the bounds in item 2 we will first need to bound the norm of the multi-variable Legendre polynomials, and the coefficients $b_{J,J'}$. The norm of the single variable Legendre polynomial is given by:

$$\int_{-1}^1 p_n(w)^2 dw = \frac{2}{2n+1}.$$

Given a multi index $J = (j_1, \ldots, j_d)$, $|J| \leq k$ we need to bound $\|p_J\|^2$. For $J = (0, \ldots, 0)$ we get that:

$$\|p_J\|^2 = 2^d.$$

For $J$ with $k$ indices that equals 1 and the rest 0 (e.g $J = (\underbrace{1, \ldots, 1}_{k}, \underbrace{0, \ldots, 0}_{d\text{-}k})$) we get that:

$$\|p_J\|^2 = \frac{2^d}{3^k}.$$

It can be easily verified that these bound the norm of the multi-variable Legendre polynomials, thus:

$$\frac{2^d}{3^k} \leq \|p_J\|^2 \leq 2^d.$$

The coefficients $b_{J,J'}$ can be calculated in the following way:

$$b_{J,J'} = \int_{w \in [-1,1]^d} w^J p_{J'}(w) dw =$$
$$\left( \int_{-1}^{1} w_1^{j_1} p_{j_1'}(w_1) dw_1 \right) \cdots \left( \int_{-1}^{1} w_d^{j_d} p_{j_d'}(w_d) dw_d \right). \tag{30}$$

Using Lemma D.1 we can bound each term in Eq. (30), and thus get a bound on $b_{J,J'}$. For any two multi-indices $J, J'$ we get the upper bound:

$$b_{J,J'} \leq 1$$

and if we assume that $|J| \leq k$, meaning that $j_1 + \cdots + j_d \leq k$, and also that $J' = J$ we get the lower bound:

$$b_{J,J} \geq 2^{-(j_1 + \cdots + j_d)} \geq 2^{-k}.$$

Now we are ready to bound the coefficients $c_J$. Denote $c_n = \max_{|J| \leq n} |c_J|$. For a multi index $J$ with $|J| = n$, we plug the bounds of $b_{J,J'}$ and $\|p_J\|^2$ into Eq. (29) to get:

$$|c_J| \leq \left( \alpha + 2^{-d} \sum_{J' < J} \frac{A}{\left(\sqrt{d}\right)^n} c_{n-1} 2^d \right) \left( 2^{-d} \frac{a}{\left(\sqrt{d}\right)^n} 2^{-n} \frac{2^d}{3^n} \right)^{-1}$$

$$\leq \frac{\alpha \left(\sqrt{d}\right)^n 6^n}{a} + \frac{A}{a} 6^n \sum_{J' < J} |c_{J'}| \leq \frac{\alpha \left(\sqrt{d}\right)^n 6^n}{a} + \frac{A}{a} 6^n \sum_{i=0}^{n-1} \binom{d}{i} c_i$$

$$\leq \frac{\alpha \left(\sqrt{d}\right)^n 6^n}{a} + \frac{A}{a} 6^n (d+1)^{n-1} c_{n-1} \leq \alpha \frac{A}{a} (12d)^n c_{n-1}, \tag{31}$$

where we used the bound $\sum_{i=1}^{n} \binom{d}{i} \leq (d+1)^n$ and that $c_i \leq c_n$ for $i \leq n$. For $J = (0, \ldots, 0)$ using Eq. (28) we get that:

$$|c_{(0,\ldots,0)}| \leq \frac{\alpha}{a}.$$

Thus using the recursion relation we found in Eq. (31) and plugging in the initial condition for $c_{(0,\ldots,0)}$ gives:

$$|c_J| \leq \alpha^n \left(\frac{A}{a}\right)^n (12d)^{n^2}$$

The final stage is to bound $g(w)$. Note that for every Legendre polynomial $\max_{|w| \leq 1} |p_n(w)| \leq 1$, hence:

$$\max_{w \in \left[\frac{-1}{\sqrt{d}}, \frac{1}{\sqrt{d}}\right]^d} |p_J\left(\sqrt{d}w\right)| \leq 1.$$

Using the definition Eq. (23) of $g(w)$, and the bound on $c_J$ we get that:

$$\max_{w \in \left[\frac{-1}{\sqrt{d}}, \frac{1}{\sqrt{d}}\right]^d} |g(w)| \leq \sum_{|J'| \leq k} |c'_J| \leq \sum_{i=0}^{k} \binom{d}{i} c_i$$

$$\leq \sum_{i=0}^{k} \binom{d}{i} c_k \leq \alpha^k \left(\frac{A}{a}\right)^k (12d)^{k^2} (d+1)^k$$

$$\leq \alpha^k \left(\frac{A}{a}\right)^k (12d)^{2k^2}.$$

$\square$

# E Random Features Concentrate Around their Expectation

*Proof of Theorem C.4.* Define $u_i = \frac{g(w_i)}{r}$, and

$$\hat{f}(w_1, \ldots, w_r, x) = \hat{f}(x) = \sum_{i=1}^{r} u_i \sigma(\langle w_i, x \rangle).$$

Observe that $\mathbb{E}_w\left[\hat{f}(x)\right] = f(x)$ and $|u_i| \leq \frac{C}{r}$. Define:

$$h(w_1, \ldots, w_r, x) = h(x) = \sup_x \left| \hat{f}(w_1, \ldots, w_r, x) - \mathbb{E}_w\left[\hat{f}(w_1, \ldots, w_r, x)\right] \right|$$

We will use McDiarmid's inequality to bound $h$. For every $1 \leq i \leq r$ and every $\widetilde{w_i}$ with $\|\widetilde{w_i}\| \leq 1$ we have that:

$$|h(w_1, \ldots, w_r, x) - h(w_1, \ldots, w_{i-1}, \widetilde{w_i}, w_{i+1}, \ldots, w_r, x| \leq$$

$$\leq \sup_x \left| \frac{g(w_i)\sigma(\langle w_i, x \rangle)}{r} - \frac{g(\widetilde{w_i})\sigma(\langle \widetilde{w_i}, x \rangle)}{r} \right| \leq \frac{2LC}{r}$$

We will now bound the expectation of $h(x)$. Using [32, Lemma 26.2] which bounds $\mathbb{E}[h(x)]$ using Rademacher complexity, where the roles of $x$ and $w$ are switched:

$$\mathbb{E}[h(x)] = \mathbb{E}\left[\sup_x \left| \hat{f}(x) - f(x) \right|\right] \leq \frac{2}{r} \mathbb{E}_{w,\xi}\left[\sup_x \left| \sum_{i=1}^{r} \xi_i u_i \sigma(\langle w_i, x \rangle) \right|\right]$$

Where $\xi_1, \ldots, \xi_r$ are independent Rademacher random variables (where we write them as $\xi$ for short). Define $\sigma'(x) = \sigma(x) - \alpha$, where $\sigma(0) = \alpha$, then we have that $\sigma'(0) = 0$. We use the fact that for i.i.d Rademacher random variables $\xi_1, \ldots, \xi_r$:

$$\mathbb{E}_\xi\left[\left| \sum_{i=1}^{r} \xi_i \right|\right] \leq \sqrt{r},$$

combined with [6, Theorem 12(4)], Cauchy-Schwartz theorem and our assumptions that $\|x\|, \|w\| \leq 1$ to get:

$$\frac{2}{r} \mathbb{E}_{w,\xi}\left[\sup_x \left| \sum_{i=1}^{r} \xi_i u_i \sigma(\langle w_i, x \rangle) \right|\right] \leq \frac{2}{r} \mathbb{E}_{w,\xi}\left[\sup_x \left| \sum_{i=1}^{r} \xi_i u_i \sigma'(\langle w_i, x \rangle) \right| + \alpha \left| \sum_{i=1}^{r} \xi_i u_i \right|\right]$$

$$\leq \frac{2LC}{r} \mathbb{E}_{w,\xi}\left[\sup_x \left| \sum_{i=1}^{r} \xi_i \langle w_i, x \rangle \right|\right] + \frac{2C}{r} \mathbb{E}_\xi\left[\alpha \left| \sum_{i=1}^{r} \xi_i \right|\right]$$

$$\leq \frac{2LC}{r} \mathbb{E}_\xi\left[\left| \sum_{i=1}^{r} \xi_i \right|\right] + \frac{2LC}{r} \mathbb{E}_\xi\left[\left| \sum_{i=1}^{r} \xi_i \right|\right]$$

$$\leq \frac{4LC}{\sqrt{r}}$$

In total we have:

$$\mathbb{E}[h(x)] \leq \frac{4LC}{\sqrt{r}}$$

We can now use McDiarmid's inequality on $h(x)$ to get that:

$$P\left(h(x) - \frac{4LC}{\sqrt{r}} \geq \epsilon\right) \leq P\left(h(x) - \mathbb{E}_w(h(x)) \geq \epsilon\right) \leq \exp\left(-\frac{r\epsilon^2}{4L^2C^2}\right) \tag{32}$$

Replacing the right hand side with $\delta$ we get that w.p $> 1 - \delta$:

$$\sup_x \left|\hat{f}(x) - f(x)\right| \leq \frac{LC}{\sqrt{r}}\left(4 + \sqrt{2\log\left(\frac{1}{\delta}\right)}\right)$$

$\square$

## F    SGD on Over-Parameterized Networks Competes with Random Features

**Lemma F.1.** *Let* $\|W_0\|$, $\|U_0\| \leq B$ *with* $B \geq 2$, *then for every* $\epsilon > 0$ *if we run SGD with learning rate of* $\eta = \frac{\epsilon}{LB^2}$ *we have that for all* $t \leq \frac{B}{2\epsilon}$:

1. $\|W_t\|, \|U_t\| \leq B + 1$

2. $\|\sigma(W_t x) - \sigma(W_0 x)\| \leq 2Lt\epsilon$

*Proof.* We prove the first part by induction on $t$. First trivially it is true for $t = 0$. Assume it is true for all $t \leq \frac{B}{2\epsilon}$. The gradients of $L_D(U, W)$ are:

$$\frac{\partial l\left(N(W,U,x),y\right)}{\partial W} = \mathbb{1}_{(1-y\cdot N(U,W,x)\geq 0)}(x\tilde{U})^T \tag{33}$$

$$\frac{\partial l(N(W,U,x),y)}{\partial U} = \mathbb{1}_{(1-y\cdot N(U,W,x))}\sigma(Wx) \tag{34}$$

Here $\tilde{U}_i = u_i \cdot \sigma'(\langle w_i, x\rangle)$, and we look at $x$ as a matrix in $\mathbb{R}^{d\times 1}$ hence $x\tilde{U} \in \mathbb{R}^{d\times r}$.

We bound the gradients of $L_D(U, W)$ using Eq. (34) and Eq. (33), the assumptions on $\sigma$ and that $\|x\| \leq 1$:

$$\left\|\frac{\partial l\left(N(W,U,x),y\right)}{\partial W}\right\| \leq L\|U\|$$

$$\left\|\frac{\partial l\left(N(W,U,x),y\right)}{\partial U}\right\| \leq \|\sigma(Wx)\| \leq \|\sigma(Wx) - \sigma(0)\| + \|\sigma(0)\| \leq L + L\|W\|$$

Using the bounds on the gradient, at each step of SGD the norm of $W_{t+1}$ changed from the norm of $W_t$ by at most $\eta L\|U_t\|$. Thus, after $t$ iterations we get that

$$\|W_{t+1}\| \leq \|W_0\| + \sum_{i=1}^{t}\eta L\|U_i\| \leq B + t\eta L(B+1) \leq B + \frac{\epsilon}{B^2}\frac{B}{2\epsilon}(B+1) \leq B+1.$$

In the same manner for $U_{t+1}$:

$$\|U_{t+1}\| \leq \|U_0\| + \sum_{i=1}^{t}\eta(L + L\|W_i\|) \leq B + t\eta L(B+1) + t\eta L$$

$$\leq B + \frac{\epsilon}{B^2}\frac{B}{2\epsilon}(B+1) + \frac{1}{2B} \leq B+1.$$

For the second part, using the previous part we get that:

$$\|W_{t+1} - W_0\| \leq \sum_{i=1}^{t}\eta\|U_i\| \leq t\eta(B+1) = t\epsilon\frac{B+1}{B}.$$

Now we use the fact that $\sigma$ is $L$-Lipschitz, $\|x\| \leq 1$ and $|B| \geq 1$ to get that:

$$\|\sigma(W_t x) - \sigma(W_0 x)\| \leq 2Lt\epsilon.$$

$\square$

We will also use the following theorem about convex online learning (see [32, Theorem 21.15]):

**Theorem F.2.** *Let $f_1, \ldots, f_T : \mathbb{R}^d \to \mathbb{R}$ be L-Lipschitz convex functions. Assume that $x_{t+1} = x_t - \eta \nabla f_t(x_t)$, then for any $x^* \in \mathbb{R}^d$ we have that:*

$$\sum_{t=1}^{T} f_t(x_t) \leq \sum_{t=1}^{T} f_t(x^*) + \frac{\|x^* - x_0\|^2}{2\eta} + \frac{\eta T L^2}{2}$$

Now we are ready to prove the generalization bound:

*Proof of Theorem C.3.* For every $1 \leq t \leq T$ we have by Lemma F.1 that:

$$|L_D(W_t, U^*) - L_D(W_0, U^*)| \leq \|U^*\| \cdot \mathbb{E}_x \left[\|\sigma(W_t x) - \sigma(W_0 x)\|\right] \leq \frac{C}{\sqrt{r}} 2Lt\epsilon \leq \epsilon \quad (35)$$

Where the last inequality is by the choice of $r$. We define the function $g_t(U) := l(N(W_t, U, x_t), y_t)$ where $(x_t, y_t)$ is the example sampled at round $t$ of SGD. Observe that:

$$|g_t(U) - g_t(U')| \leq \|\sigma(W_t x)\| \cdot \|U - U'\|$$

where we used the fact that the loss is 1-Lipschitz. Also note that $g_t(U)$ are convex for every $t$. Using Lemma F.1 again:

$$|\sigma(W_t x)| \leq |\sigma(W_t x) - \sigma(0)| + |\sigma(0)| \leq L(\sqrt{r} + 1) + L \leq 2L\sqrt{r}$$

thus $g_t(U)$ is also $2L\sqrt{r}$-Lipschitz for all $t$. We use Theorem F.2 on the functions $g_t$ to get:

$$\sum_{t=1}^{T} g_t(U_t) \leq \sum_{t=1}^{T} g_t(U^*) + \frac{\|U^* - U_0\|^2}{2\eta} + 8\eta r T L^2 \quad (36)$$

Dividing by $T$ we get that:

$$\frac{1}{T} \sum_{t=1}^{T} g_t(U_t) \leq \frac{1}{T} \sum_{t=1}^{T} g_t(U^*) + \frac{\|U^* - U_0\|^2}{2\eta T} + 8\eta r \quad (37)$$

Using the lower bound of $T$ we get that:

$$\frac{\|U^* - U_0\|^2}{2\eta T} \leq \frac{\|U^*\|^2}{2\eta T} + \frac{\|U_0\|^2}{2\eta T} \leq 2\epsilon \quad (38)$$

Combining Eq. (37) with Eq. (38) and plugging in $\eta$ gives us:

$$\frac{1}{T} \sum_{t=1}^{T} g_t(U_t) \leq \frac{1}{T} \sum_{t=1}^{T} g_t(U^*) + 3\epsilon \quad (39)$$

Observe that taking expectation of $g_t$ with respect to the sampled examples in round $t$ of SGD yields: $\mathbb{E}[g_t(U)] = L_D(W_t, U)$. Thus, taking expectation on Eq. (39) and using Eq. (35):

$$\frac{1}{T} \sum_{t=1}^{T} \mathbb{E}[L_D(W_t, U_t)] \leq L_D(W_0, U^*) + 4\epsilon \quad (40)$$

Thus there is $1 \leq t \leq T$ that satisfies:

$$\mathbb{E}[L_D(W_t, U_t)] \leq L_D(W_0, U^*) + 4\epsilon$$

Rescaling $\epsilon$ appropriately finishes the proof. $\qquad \square$

# G   Approximating polynomials with ReLU networks

Theorem C.1 can be modified to also include the ReLU activation which is not analytic. This modification requires to add a bias term and also use a non-standard architecture for the network. For terseness we explain here how it can be done without writing the full proof:

We begin with the following network architecture:

$$N(W, U, b) = \sum_{i=1}^{r} u_i[\langle w_i, x \rangle - b_i]_+ - u_i[\langle -w_i, x \rangle - b_i]_+ + cu_i \cdot \langle w_i, x \rangle + cu_i,$$

where $c = \frac{1}{e-1}$ is a normalization term which is added for simplicity. This architecture is similar to a standard feed-forward neural network, but includes duplicated ReLU neurons with a negative sign, and linear and constant factors. The initialization of $w_i$ and $u_i$ is the same as in Theorem C.1, and the bias terms $b_i$ are initialized from a uniform distribution on $[0, 1]$.

Steps 1 and 2 are similar to those used in the original theorem, with adjustments for the added terms, and also in step 2 the function $g(w)$ should depend additionally on the bias term $g(w, b)$. Thus, we can approximate an integral of the form:

$$\int_{w \in \left[\frac{-1}{\sqrt{d}}, \frac{1}{\sqrt{d}}\right]^d} \int_0^1 g(w, b)[\langle w, x \rangle - b]_+ - g(w, b)[\langle -w, x \rangle - b]_+ + cg(w, b)\langle w, x \rangle + cg(w, b) \, db \, dw. \tag{41}$$

For any $z \in \mathbb{R}$ with $|z| \le 1$ we have that:

$$\int_0^1 [z - b]_+ e^b - [-z - b]_+ e^{-b} + cze^b + ce^b \, db = e^z \tag{42}$$

Plugging in $g(w, b)$ into Eq. (41), and using the integral from Eq. (42) with $z = \langle w_i, x \rangle$ (note that $|\langle w_i, x \rangle|^2 \le \|w\|^2 \|x\|^2 \le 1$) we can approximate an integral of the form:

$$\int_{w \in \left[\frac{-1}{\sqrt{d}}, \frac{1}{\sqrt{d}}\right]^d} g(w) \exp(\langle w, x \rangle) \, dw.$$

Now we can use step 3 to finish the proof.

The requirement for the extra linear and constant terms are also needed in [21]. There it is shown that functions that admits certain Fourier transform representations can be approximated using a combination of ReLUs, with an extra linear and constant factors.

## Footnotes

[1]In order to deduce it we only need to note that since $\psi$ is odd, its first Fourier coefficient is $a_0 = 0$, and we can take $g = \widehat{f\varphi}$ and use the fact that Fourier transform preserves inner products and norms.