[Reviews · NeurIPS 2019]

Reviewer 1



Originality? This work is the first to concretely show that random features alone cannot be used to explain why overparameterized networks work well. This work is introduces a healthy amount of skepticism in employing vague intuitions for explaining "why neural networks work." This is neither the first nor the last paper with such a premise, but it is a valuable addition to the community. Quality? The mathematical foundations of the paper appear solid. I did not check the math thoroughly. Clarity? I find this work moderately unclear. Explicitly introducing the key theorems, their consequences, and their limitations early on would go a long way to framing the rest of the paper. Significance? I find the significance of this work low yet valuable. This paper illustrates a particular example dataset that is difficult to learn with random features. The existence of such a dataset doesn't necessarily imply the "random features" interpretaion of neural networks is invalid; simply that it is insufficient on its own.

Reviewer 2



*** Comments after rebuttal *** Even though the authors have answered some of my concerns, the main issue that leads me to keep my original score and recommend rejection is that it is not clear if the clarity and correctness of the proof will be satisfactory after the suggested rewriting. This would require a new review that cannot be performed in the current conference cycle. Regarding your answer to 3b,c this notation is really misleading if you want to mean the norm of a function then you should use || f || not || f(x) || *** Original review *** Originality. The authors focus on negative results that show some fundamental limits on what can be learned with random features, which tend to deviate from previous works that rather aim to provide positive results. Understanding the limits of random features is a valuable avenue of research. However I find that the included positive result (Theorem 3.1) is difficult to assess as a significant contribution as even the authors claim that it is not fundamentally novel and is a rather new/more compact/better proof than that of other similar results. Quality. I find there are some major flaws in some parts, or either I am misunderstanding something. It is difficult to assess a theory-only paper in a conference cycle and I have tried my best to understand the contribution and review the proofs. (1) the main claim is that random features "cannot be used to learn even a single neuron", however the theoretical result is that the product between the number of random features 'r' and the maximum of the linear combination coefficients 'u_i' is exponential in the dimension. Even though it is clear that an exponential number of required random features means the impossibility result, I still cannot grasp why exponential coefficients in the linear combination also mean that one cannot learn. The authors seem to write such result and not discuss why that implies the negative result. As remark 4.3. points out the authors believe the dependence on the 'u_i' can be removed, and that would certainly be a better result. (2) Theorem 2 has a condition where d > c_1 the dimension should be larger than some constant but I didn't see any discussion on how large c_1 can be, so that one could grasp if the results apply to common dimensionality magnitudes in current applications. It would not be so significant if the lower bound c_1 is something unrealistic and there doesnt seem to be any discussion around this. (3) In the proof of theorem 4.2, line 274 reads "Assume that for every j we can find u_j in R^r and f_1, ..., f_r such that ... " It is not clear if this assumption comes from the statement of the theorem, or if it is used as an intermediate result it should be written as a proposition. If some assumption is included it should be stated why (intermediate result, to obtain a contradiction, split by cases) otherwise what happened in my case is that I cannot understand the general structure of the proof and this is something bad when the main contribution is theoretical. Clarity. The paper reads ok in the prose sections but I found some problems in the proof of theorem 4.2 (as stated in the previous Quality section) and the following: (1) line 123 "a key element is to analyze neural networks as if they are random features either explicitely or implicitely", this phrase is too hard to understand and grasp what the authors mean. (2) section 3 feels not relevant when the authors say that 'it improves on certain aspects ... discussed in appendix' which means that the improvements are not clear or minor, and when they say 'the analysis ... is not fundamentally novel', so it feels that this sections just fills some space. (3) Proposition 4.4. is not properly written as it is not clear if the statements are about the function \psi or the value \psi(x), for which no quantifier is introduced (all x? some x?) for example (3a) line 247 says 'it is periodic function', so it should refer to the function \psi not \psi(x), (3b) line 249: there is an x in the expression but there is no quantifier for such value (any x? some x?) and the same for item 3. The absence of quantifiers makes the proposition hard to read. It should be correctly rewritten. (4) Finally the paper ends abruptly and leaves the reader confused about the whole point of the paper, In my opinion there should be a conclusion and future work section, wrapping up the results presented, their limitations and why are they important, and the directions of future work and why are they relevant. Significance. For me it is too hard to assess how significant this results are given what I explained in the quality and clarity sections. I think it is valuable to have negative results on the capabilities of learning with random features, but the authors should explain a little bit better why their results imply this impossibility (in particular why the exponential magnitudes of the coefficient u_i woudl imply this, maybe I am missing something)

Reviewer 3



In this article, the authors studied the limitation of the random feature-based single-hidden-layer neural network model. By considering a standard d-dimensional Gaussian model for the input data, the authors showed that, for a large class of activation function f, the random feature transformation of the type u' f(W x) for (the rows of) W uniformly distributed on the unit sphere cannot well approximate a single ReLU neuron output, unless either the number of random features or the amplitude of the second layer weight u are allowed to grow exponentially with respect to d. This article brings novel understanding into random feature-based techniques, and perhaps more importantly, provides the critical analysis arguing that, despite their nice theoretical properties, it seems unlikely that random feature techniques are behind the many success of deep neural networks today. I am pleased to recommend it for publication. **After rebuttal**: I have read the author response and my score remains the same.

Reviewer 4



The main lower bound on random features approximating RELU is quite nice and elegant. It builds on the idea that random features cannot approximate certain oscillating functions (which is implicit in some previous work) which are approximable by RELUs. The authors then use an elegant symmetrization argument to say that if your random features are coming from a nice distribution, then you should not be able to approximate even a single RELU. The argument to rule out more general classes of random features is similar in spirit. In comparison, it is known that a single RELU (under the distribution class for which the above lower bound holds) can be learned as a RELU by standard gradient descent methods. Regarding significance, the paper presents a strong case that the line of thought based on random features and over-parametrization in several recent works needs new ideas (at a conceptual level) to shed light on the effectiveness of deep learning. Comments on writeup: By and large the paper is quite well-written but there were a few statements that seemed vague. It will be helpful to clarify these parts more. 1. Line 63/64 - the sentence needs rephrasing (starts with random features and then has random or deterministic in parentheses). 2. Line 72: 'vanilla neural networks' is too vague at this point. 3. 75,76: 'easily learnable' is also a bit vague. It is indeed early in the paper so perhaps adding a pointer to a later remark explaining this would be helpful. 4. 172-176: It would be very helpful for readers (in assessing the literature) if you can point out which of the previous works fall into which category (of the four you listed). Else, chasing them down for clarity would be difficult.

[Author Response · NeurIPS 2019]

We thank all reviewers for their helpful and detailed comments.

Review 1

Regarding the improvement suggestions:
- We will present the main results and limitations more explicit in the introduction section of the paper.

Review 2

Regarding the remarks under the "quality" bullet:
(1) We will add a discussion on the requirement that the $u_i$ coefficients are exponentially large. In a nutshell, existing
analyses of stochastic gradient descent, even for convex functions, imply that the required number of iterations scales
polynomially with the norm of the target solution, which would mean exponentially many iterations in our case.
Moreover, practically speaking, such huge coefficients can cause overflow when running SGD on a computer with
standard floating point formats.
(2) $c_1$ is a small numerical constant that does not depend on any parameter of the theorem (it comes from Lemma 17 in
[1], quantifies concentration of measure on a sphere, and can be explicitly upper bounded by $40$). We will try to make
this clearer.
(3) This is a proof by contradiction, and this assumption is what we want to show to be invalid. We will write this in a
clearer way.

Regarding the remarks under the "clarity" bullet:
(1) We agree that "explicitly or implicitly" is not sufficiently clear, and we will rephrase. What we meant is simply that
all the papers discussed in section 2 use the random features idea in various ways.
(2) The goal of section 3 is to give a simple self-contained proof on how neural networks can be explained using neural
networks, and to give motivation to the forthcoming section. As we explicitly point out, the proof methods are not
that novel, which is why this section is only about half a page (although we do improve on previous results regarding
approximations of polynomials).
(3) We will rephrase this notation to make it clearer. It should be written that (3a) $\psi : \mathbb{R} \to \mathbb{R}$ is a real periodic function.
(3b,c) It is the norm defined at the beginning of the section (lines 199-200).
(4) We will try to add a conclusions section in the final version (appropriate to the page limit).

Review 3

Regarding the improvement suggestions:
- Regarding the "uniformly spherically distributed" assumption on $W$: The theorem can be readily extended so that $W$
has a standard Gaussian distribution, though it would require more complicated calculations as we would need to bound
the norm of the function $f_W$ w.h.p, instead of an absolute bound which we used in the theorem. We prefer to keep the
theorem that way to make the proofs easier to understand. However, we can add a comment if the reviewer feels it is
needed.
- In the relevant theorems, we will make it clearer when $\mathbf{x}$ is assumed to have a Gaussian distribution.
- Regarding extension to the "linearized" neural tangent kernel model: In fact, Theorem 4.6 applies to this model (it
does not make any specific assumptions on the feature class $\mathcal{F}$). We will add an explicit comment on that.
- We will fix the boldface notation where relevant.

**References**

[1] Shamir, Ohad. "Distribution-specific hardness of learning neural networks." The Journal of Machine Learning
Research 19.1 (2018): 1135-1163.


[Meta-Review · NeurIPS 2019]

This paper shows that random feature methods can not efficiently learn even a single ReLU. This has stark implications for a lot of recent work trying to explain the success of deep learning through random feature methods. The authors however should take into account the reviews and improve the writing and presentation.